# ON THE REGULARIZATION OF WASSERSTEIN GANS

**Henning Petzka**[*]
Fraunhofer Institute IAIS,
Sankt Augustin, Germany
`henning.petzka@gmail.com`

**Asja Fischer**[*]**& Denis Lukovnikov**
Department of Computer Science,
University of Bonn, Germany
`asja.fischer@gmail.com`
`lukovnik@cs.uni-bonn.de`

## ABSTRACT

Since their invention, generative adversarial networks (GANs) have become a popular approach for learning to model a distribution of real (unlabeled) data. Convergence problems during training are overcome by Wasserstein GANs which minimize the distance between the model and the empirical distribution in terms of a different metric, but thereby introduce a Lipschitz constraint into the optimization problem. A simple way to enforce the Lipschitz constraint on the class of functions, which can be modeled by the neural network, is weight clipping. Augmenting the loss by a regularization term that penalizes the deviation of the gradient norm of the critic (as a function of the network's input) from one, was proposed as an alternative that improves training. We present theoretical arguments why using a weaker regularization term enforcing the Lipschitz constraint is preferable. These arguments are supported by experimental results on several data sets.

## 1 INTRODUCTION

General adversarial networks (GANs) (Goodfellow et al., 2014) are a class of generative models that have recently gained a lot of attention. They are based on the idea of defining a game between two competing neural networks (NNs): a generator and a classifier (or discriminator). While the classifier aims at distinguishing generated from real data, the generator tries to generate samples which the classifier can not distinguish from the ones from the empirical distribution. Realizing the potential behind this new approach to generative models, more recent contributions focused on the stabilization of training, including ensemble methods (Tolstikhin et al., 2017), improved network structure (Radford et al., 2015; Salimans et al., 2016) and theoretical improvements (Nowozin et al., 2016; Salimans et al., 2016; Arjovsky & Bottou, 2017; Chen et al., 2016) that helped to successfully model complex distributions using GANs.

It was proposed by Arjovsky et al. (2017) to train generator and discriminator networks by minimizing the Wasserstein-1 distance, a distance with properties superior to the Jensen-Shannon distance (used in the original GAN) in terms of convergence. Accordingly, this version of GAN was called Wasserstein GAN (WGAN). The change of metric introduces a new minimization problem, which requires the discriminator function to lie in the space of 1-Lipschitz functions. In the same paper, the Lipschitz constraint was guaranteed by performing weight clipping, i.e., by constraining the parameters of the discriminator NN to be smaller than a given value in magnitude. An improved training strategy was proposed by Gulrajani et al. (2017) based on results from optimal transport theory (see Villani, 2008). Here, instead of clipping weights, the loss gets augmented by a regularization term that penalizes any deviation of the norm of the gradient of the critic function (with respect to its input) from one.

We review these results and present both theoretical considerations and empirical results, leading to the proposal of a less restrictive regularization term for WGANs.[1] More precisely, our contributions are as follows:

---

[*]Equal contributions

[1]In the blog post `https://lernapparat.de/improved-wasserstein-gan/` which was written simultaneously to our work, the author presents some ideas that follow a similar intuition as the one underlying our arguments.

- We review the arguments that the regularization technique proposed by Gulrajani et al. (2017) is based on and make the following two observations: (i) The regularization strategy requires training samples and generated samples to be drawn from a certain joint distribution. In practice, however, samples are drawn independently from their marginals. (ii) The arguments further assume the discriminator to be differentiable. We explain why both can be harmful for training.

- We propose a less restrictive regularization term and present empirical results strongly supporting our theoretical considerations.

## 2  OPTIMAL TRANSPORT

We will require the notion of a coupling of two probability distributions. Although a coupling can be defined more generally, we state the definition in the setting of our interest, i.e., we consider all spaces involved to equal $\mathbb{R}^n$.

**Definition 1.** *Let $\mu$ and $\nu$ be two probability distributions on $\mathbb{R}^n$. A coupling $\pi$ of $\mu$ and $\nu$ is a probability distribution on $\mathbb{R}^n \times \mathbb{R}^n$ such that $\pi(A, \mathbb{R}^n) = \mu(A)$ and $\pi(\mathbb{R}^n, A) = \nu(A)$ for all measurable sets $A \subseteq \mathbb{R}^n$. The set of all couplings of $\mu$ and $\nu$ is denoted by $\Pi(\mu, \nu)$.*

The following theorem plays a central role in the theory of optimal transport (OT) and is known as the Kantorovich duality. Note, that the presented theorem is a less general, but to our needs adapted version of Theorem 5.10 from Villani (2008).[2] A proof of how to derive our version from the referenced one can be found in Appendix C.1. We will denote by $\mathcal{L}ip_1$ the set of all 1-Lipschitz functions, i.e., the set of all functions $f$ such that $f(y) - f(x) \leq ||x - y||_2$ for all $x, y$.

**Theorem 1** (Kantorovich). *Let $\mu$ and $\nu$ be two probability distributions on $\mathbb{R}^n$ such that $\int_{\mathbb{R}^n} ||x||_2 \, d\mu(x) < \infty$ and $\int_{\mathbb{R}^n} ||x||_2 \, d\nu(x) < \infty$. Then*

*(i)*
$$\min_{\pi \in \Pi(\mu, \nu)} \int_{\mathbb{R}^n \times \mathbb{R}^n} ||x - y||_2 \, d\pi(x, y) = \max_{f \in \mathcal{L}ip_1} \left( \int_{\mathbb{R}^n} f(x) \, d\mu(x) - \int_{\mathbb{R}^n} f(x) \, d\nu(x) \right) \ . \quad (1)$$

*In particular, both minimum and maximum exist.*

*(ii) The following two statements are equivalent:*

*(a) $\pi^*$ is an optimal coupling (minimizing the value on the left hand side of (1)).*

*(b) Any optimal function $f^* \in \mathcal{L}ip_1$ (at which the maximum is attained for the right hand side of (1)) satisfies that for all $(x, y)$ in the support of $\pi^*$: $f^*(x) - f^*(y) = ||x - y||_2$.*

The field of OT offers several approaches to the computation of optimal couplings. To speed up the computation of an optimal coupling, Cuturi (2013) introduced a regularized version of the primal problem in which an entropic term $E(\pi)$ is added leading to the minimization of $\int_{\mathbb{R}^n \times \mathbb{R}^n} ||x - y||_2 \, d\pi(x, y) + \epsilon E(\pi)$, with regularization parameter $\epsilon$. Regularized OT was generalized by Dessein et al. (2016) to a more general class of regularization terms $\Omega(\pi)$. As we will discuss in Section 5, the learning algorithm we propose in this paper has connections to the approach using $\Omega(\pi) = \int \left( \frac{d\pi(x, y)}{d\mu(x) \, d\nu(y)} \right)^2 d\mu(x) \, d\nu(y)$. By Blondel et al. (2017), this leads to the dual problem given by

$$\sup_{f, g} \left\{ \mathbb{E}_{x \sim \mu}[f(x)] - \mathbb{E}_{y \sim \nu}[g(y)] - \frac{4}{\epsilon} \int \int \max \{0, (f(x) - g(y) - ||x - y||_2)\}^2 \, d\mu(x) d\nu(y) \right\} \ . \quad (2)$$

## 3  WASSERSTEIN GANS

Formally, given an empirical distribution $\mu$, a class of generative distributions $\nu$ over some space $\mathcal{X}$, and a class of discriminators $d : \mathcal{X} \to [0, 1]$, GAN training (Goodfellow et al., 2014) aims at

---

[2]In the work of Arjovsky et al. (2017), the theorem is called Kantorovich-Rubinstein, although this theorem only applies to compact metric spaces. There are several generalizations of the Kantorovich-duality. For a detailed account we refer the reader to Edwards (2011).

solving the optimization problem given by $\min_\nu \max_d \mathbb{E}_{x \sim \mu}[\log(d(x))] + \mathbb{E}_{y \sim \nu}[\log(1 - d(y))]$.[3] In practice, the parameters of the generator and the discriminator networks are updated in an alternating fashion based on (several steps) of stochastic gradient descent. The discriminator thereby tries to assign a value close to zero to generated data points and values close to one to real data points. As an opposing agent, the generator aims to produce data where the discriminator expects to see real data. Theorem 1 by Goodfellow et al. (2014) shows that, if the optimal discriminator is found in each iteration, minimization of the resulting loss function of the generator leads to minimization of the *Jensen-Shannon* (JS) divergence. Instead of minimizing the JS divergence, Arjovsky et al. (2017) proposed to minimize the *Wasserstein-1* distance, also known as *Earth-Mover* (EM) distance, which is defined for any Polish space $(M, c)$ and probability distributions $\mu$ and $\nu$ on $M$ by

$$W(\mu, \nu) = \inf_{\pi \in \Pi(\mu,\nu)} \int_{M \times M} c(x,y) \, d\pi(x,y) \ . \tag{3}$$

From the Kantorovich duality (see Theorem 1, (i)) it follows that, in the special case we are considering, the infimum is attained and, instead of computing this minimum in Equation (3), the Wasserstein-1 distance can also be computed as

$$W(\mu, \nu) = \max_{f \in \mathcal{L}ip_1} \mathbb{E}_{x \sim \mu}[f(x)] - \mathbb{E}_{y \sim \nu}[f(y)] \ , \tag{4}$$

where the maximum is taken over the set of all 1-Lipschitz functions $\mathcal{L}ip_1$.

Thus, the WGAN objective is to solve

$$\min_\nu \max_{f \in \mathcal{L}ip_1} \mathbb{E}_{x \sim \mu}[f(x)] - \mathbb{E}_{y \sim \nu}[f(y)] \ , \tag{5}$$

which can be achieved by alternating gradient descent updates for the generating network $\nu$ and the 1-Lipschitz function $f$ (also modeled by a NN), just as in the case of the original GAN. The objective of the generator is still to generate real-looking data points and is led by function values of $f$ that plays the role of an appraiser (or critic). The appraiser's goal is to assign a value of confidence to each data point, which is as low as possible on generated data points and as high as possible on real data. The confidence value it can assign is bounded by a constraint of similarity, where similarity is measured by the distance of data points. This can be motivated by the idea that similar points should have similar values of confidence for being real. The new role of the critic helps to solve convergence problems, but the interpretation of its value as classifying real (close to 1) and fake data (close to 0) is lost. We refer to Appendix A for a detailed discussion.

## 4 IMPROVED TRAINING OF WGANS

Modeling the WGAN critic function by a NN raises the question on how to enforce the 1-Lipschitz constraint of the objective in Equation (5). As proposed by Arjovsky et al. (2017) a simple way to restrict the class of functions $f$ that can be modeled by the NN to $\alpha$-Lipschitz continuous functions (for some $\alpha$) is to perform weight clipping, i.e. to enforce the parameters of the network not to exceed a certain value $c_{\max} > 0$ in absolute value. As the authors note, this is not a good but simple choice. We further demonstrate this in Appendix B by proving (for a standard NN architecture) that, using weight clipping, the optimal function is in general not contained in the class of functions modeled by the network.

Recently, an alternative to weight clipping was proposed by Gulrajani et al. (2017). The basic idea is to augment the WGAN loss by a regularization term that penalizes the deviation of the gradient norm of the critic with respect to its input from one (leading to a variant referred to as WGAN-GP, where GP stands for gradient penalty). More precisely, the loss of the critic to be minimized is then given by

$$\mathbb{E}_{y \sim \nu}[f(y)] - \mathbb{E}_{x \sim \mu}[f(x)] + \lambda \mathbb{E}_{\hat{x} \sim \tau}[(||\nabla f(\hat{x})||_2 - 1)^2] \ , \tag{6}$$

where $\tau$ is the distribution of $\hat{x} = tx + (1 - t)y$ for $t \sim U[0, 1]$ and $x \sim \mu$, $y \sim \nu$ being a real and a generated sample, respectively. The regularization term is derived based on the following result.

---

[3]Usually, both the generative distribution and the discriminator are modeled by NNs, whose structure determine the two classes we are optimizing over.

**Proposition 1.** *Let $\mu$ and $\nu$ be two probability distributions on $\mathbb{R}^n$. Let $f^*$ be an optical critic, leading to the maximum $\max_{f \in \mathcal{L}ip_1} \int_{\mathbb{R}^n} f(x) \, d\mu(x) - \int_{\mathbb{R}^n} f(x) \, d\nu(x)$ , and let $\pi^*$ be an optimal coupling with respect to $\min_{\pi \in \Pi(\mu,\nu)} \int_{\mathbb{R}^n \times \mathbb{R}^n} ||x - y||_2 \, d\pi(x,y)$ . If $f^*$ is differentiable and $x_t = tx + (1-t)y$ for $0 \le t \le 1$, it holds that $\mathbb{P}_{(x,y)\sim\pi^*}\left[(\nabla f^*(x_t) = \frac{y - x_t}{||y - x_t||})\right] = 1$ . This in particular implies, that the norms of the gradients are one $\pi^*$-almost surely on such points $x_t$.*

For the convenience of the reader, we provide a simple argument for obtaining this result in Appendix C.2.

Note, that Proposition 1 holds only when $f^*$ is differentiable and $x$ and $y$ are sampled from the optimal coupling $\pi^*$. However, sampling independently from the marginal distributions $\mu$ and $\nu$ very likely results in points $(x, y)$ that lie outside the support of $\pi^*$. Furthermore, the optimal cost function $f^*$ does not need not to be differentiable everywhere. These two points will be discussed in more detail in the following subsections.

## 4.1 SAMPLING FROM THE MARGINALS INSTEAD OF THE OPTIMAL COUPLING

**Observation 1.** *Suppose $f^* \in \mathcal{L}ip_1$ is an optimal critic function and $\pi^*$ the optimal coupling determined by the Kantorovich duality in Theorem 1. Then $|f^*(y) - f^*(x_t)| = ||x_t - y||_2$ on the line $x_t = tx + (1-t)y$, $0 \le t \le 1$, for $(x, y)$ sampled from $\pi^*$, but not necessarily on the lines connecting an arbitrary pair of a real and a generated data point, i.e. arbitrary $x \sim \mu$ and $y \sim \nu$.*

Consider the examples in Figure 1, where every X denotes a sample from the generator and every O a real data sample. Optimal couplings $\pi^*$ are indicated in red, and values of an optimal critic function are indicated in blue (optimality is shown in Appendix A.1).

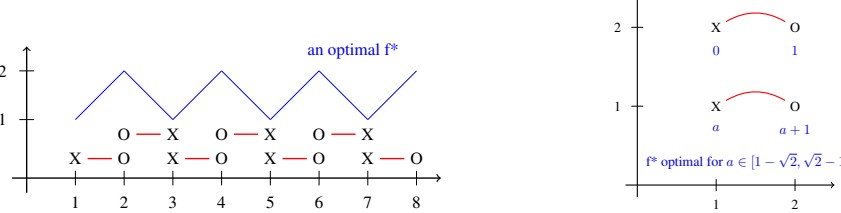

Figure 1: A one (left) and a two (right) dimensional example showing that $f^*(O)$-$f^*(X)$=|O-X| only holds for coupled pairs $(X,O) \sim \pi^*$.

In the one-dimensional example on the left, the left-most X and the right-most O satisfy $f^*(O) - f^*(X) = \frac{1}{7}|O - X| \neq |O - X|$, illustrating that the basis for the derivation of the condition, that the norm of the gradient equals one between generated and real points, only holds for points sampled from the optimal coupling. Note, while here the gradient is still of norm 1 almost everywhere, this does not necessarily hold in higher dimensions, where not all points lie on a line between some pair of points sampled from $\pi^*$. This is exemplified for two dimensions on the right side of Figure 1, where blue numbers with $a \in \mathbb{R}$ denote the values of an optimal critic function at these points (the values at these points is all that matters). Without loss of generality we can assume the value at position $(1, 2)$ to be zero, taking into account that an optimal critic function remains optimal under addition of an arbitrary constant. Since the Lipschitz constraint of $f^*$ must be satisfied, we get $1 - a \le \sqrt{2}$ and $a + 1 \le \sqrt{2}$. Therefore $a \in [1 - \sqrt{2}, \sqrt{2} - 1]$ and one of the inequalities of the Lipschitz constraint must be strict.

## 4.2 DIFFERENTIABILITY OF THE CRITIC

**Observation 2.** *The assumption of differentiability of the optimal critic is not valid at points of interest.*

Consider the example of two discrete probability distributions and its optimal critic function $f^*$ shown on the left in Figure 2. We can see that the indicated function $f^*(x) = 1 - |x| \in \mathcal{L}ip_1$

is optimal as it leads to an equality in the equation of the Kantorovich dual. (Also, it is the only continuous function, up to a constant, that realizes $f^*(x) - f^*(y) = |y - x|$ for coupled points $(x, y)$.) However, it is not differentiable at 0.

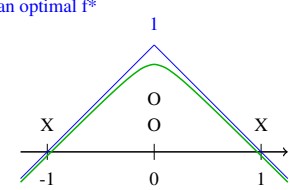 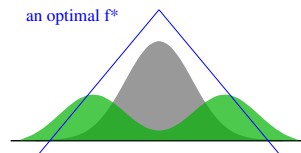

Figure 2: Non-differentiable optimal critic functions f* (shown in blue). Left: For two discrete distributions: Circles and crosses belong to samples from the empirical distribution and the generative model, respectively. An approximating differentiable function is shown in green. Right: For two continuous distributions: The empirical distribution $\mu$ is shown in gray, the generative distribution $\nu$ is shown in green.

The counterexample can be made continuous by considering the points as the center points of Gaussians, as illustrated on the right in Figure 2. This is formalized by the following proposition showing that the critic indicated in blue is indeed optimal for the depicted gray Gaussian of real data and the green mixture of two Gaussians of generated data.

**Proposition 2.** *Let $\mu = \mathcal{N}(0, 1)$ be a normal distribution centered around zero and $\nu = \nu_{-1} + \nu_1$ be a mixture of the two normal distributions $\nu_{-1} = \frac{1}{2}\mathcal{N}(-1, 1)$ and $\nu_1 = \frac{1}{2}\mathcal{N}(1, 1)$ over the real line. If $\mu$ describes the distribution of real data and $\nu$ describes the distribution of the generative model, then an optimal critic function is given by $\phi^*(x) = -|x|$.*

The proof can be found in Appendix C.3.

The issue with non-differentiability can be generalized to higher-dimensional spaces based on the observation that an optimal coupling is in general not deterministic. Deterministic couplings are particularly nice in the sense that they allow a transport plan assigning each point $x$ from one distribution deterministically to a point $y$ of the other distribution, without having to split any masses (the search for deterministic optimal couplings is called the Monge problem). However, in a lot of settings no deterministic coupling exists. The notion of a deterministic coupling is formalized in the following definition.

**Definition 2.** *Let $(X, \mu)$ and $(Y, \nu)$ be two probability spaces. A coupling $\pi \in \Pi(\mu, \nu)$ is called deterministic if there is a measurable function $\rho : X \to Y$ such that $supp(\pi) \subseteq \{(x, \rho(x)) \mid x \in X\}$.*

We can now formulate the following observation.

**Observation 3.** *Suppose $\pi^*$ is a non-deterministic optimal coupling between two probability distributions over $\mathbb{R}^n$ so that there exist points $(x, y)$ and $(x, y')$ in $supp(\pi^*)$. Suppose further that there is no $\lambda > 0$ with $(y - x) = \lambda \cdot (y' - x)$ (in particular this implies $y \neq y'$). Then any optimal critic function $f^*$ is not differentiable at $x$.*

The arguments can be found in Appendix C.5.

In practice, where the optimal critic is approximated by a NN, the situation is slightly different: A function modeled by an NN is (almost) everywhere differentiable (depending on the activation functions). By the Stone-Weierstrass theorem, on compact sets, we can approximate any (Lipschitz-)continuous function by differentiable functions uniformly. Nevertheless, it seems to be a strong constraint on an approximating function to have a gradient of norm one in the neighborhood of a non-differentiability (cf. Figure 2 (a)). Therefore, we argue – in contrast to the argumentation of Gulrajani et al. (2017) – that the gradient should not be assumed to equal one for arbitrary points on the line between an arbitrary real point $x$ and a generated point $y$.

## 5 HOW TO REGULARIZE WGANS

In the following, we will discuss how the regularization of WGANs can be improved.

**Penalizing the violation of the Lipschitz constraint.**   For the critic function, we have nothing more at hand than the inequality of the Lipschitz-constraint. Moreover (as shown in Lemma 1 in the Appendix) the exhaustion of the Lipschitz constant is automatic by maximizing the objective function. Therefore, a natural choice of regularization is to penalize the given constraint directly, i.e., sample two points $x \sim \mu$ and $y \sim \nu$ from the empirical and the generated distribution respectively and add the regularization term

$$\left( \max \left\{ 0, \frac{|f(x) - f(y)|}{||x - y||_2} - 1 \right\} \right)^2 \tag{7}$$

to the cost function. (We square to penalize larger deviations more than smaller ones.) Note the similarity of the regularization term to the squared Hinge loss, which is also used to turn a hard constraint into a soft one in the optimization problem connected to support vector machines.

Alternatively, since the NN generates (almost everywhere) differentiable functions, we can penalize whenever gradient norms are strictly larger than one, an option referred to as "one-sided penalty" and shortly discussed as an alternative to penalizing any deviation from one by Gulrajani et al. (2017)[4]. Note that enforcing the gradient to be smaller than one in norm has the advantage that we penalize when the partial derivative has norm $> 1$ into the direction of steepest descent. Hence, all partial derivatives are implicitly enforced to be bounded in norm by one, too. At the same time, enforcing $\leq 1$ for the gradient of smooth approximating functions is not an unreasonable constraint even at points of non-differentiability. For these reasons we suggest to add the regularization term $\left( \max \left\{ 0, ||\nabla f(\hat{x})|| - 1 \right\} \right)^2$ to the cost function. Different ways of sampling the point $\hat{x}$ are analyzed in Appendix D.4. Thus, our proposed method (WGAN-LP, where LP stands for Lipschitz penalty) alternates between updating the discriminator to minimize

$$\mathbb{E}_{y \sim \nu}[f(y)] - \mathbb{E}_{x \sim \mu}[f(x)] + \lambda \mathbb{E}_{\hat{x} \sim \tau}[(\max \{0, ||\nabla f(\hat{x})|| - 1\})^2] , \tag{8}$$

(where $\tau$ depends on the concrete sampling strategy chosen) and updating the generator network modeling $\nu$ to minimize $-\mathbb{E}_{y \sim \nu}[f(y)]$ using gradient descent.

**The connection to regularized optimal transport.**   Consider Equation (2) of regularized OT. For a hard constraint $f(x) - g(y) \leq ||x - y||_2$, one can attain the supremum over $\mathbb{E}_{x \sim \mu}[f(x)] - \mathbb{E}_{y \sim \nu}[g(y)] - 0$ by setting $f(x) = \inf_y g(y) + ||x - y||_2 = g(x)$ and subsequently maximize over one function only. Taking the advantage of dealing with a single function as a motivation, one may similarly replace $f = g$ in Equation 2, which uses a soft constraint (even though this can now only approximate the supremum). This leads to an objective of minimizing

$$\mathbb{E}_{y \sim \nu}[f(y)] - \mathbb{E}_{x \sim \mu}[f(x)] + \frac{4}{\epsilon} \int \int \max \{0, (f(x) - f(y) - ||x - y||_2)\}^2 \, \mathrm{d}\mu(x)\mathrm{d}\nu(y) \tag{9}$$

that, similarly to Equation (7), softly penalizes whenever $f(x) - f(y) > ||x - y||_2$ for a real sample $x$ and a generated sample $y$. It is noteworthy that to justify the replacement $f = g$ one would require a high regularization parameter $\lambda = \frac{4}{\epsilon}$ of the dual problem, which corresponds to a low regularization of the primal problem.

**Dependence on the regularization hyperparameter $\lambda$.**   Let $\mathcal{L}_\lambda^{GP}$ and $\mathcal{L}_\lambda^{LP}$ denote the infimums of the regularized losses over a class of (differentiable) critic functions $f$ from Equation (6) (WGAN-GP) and Equation (8) (WGAN-LP) respectively. For the comparison of these optimal losses we have the following result (proof in Appendix C.4).

**Proposition 3.**
$$\mathcal{L}_\lambda^{LP} \leq \mathcal{L}_\lambda^{GP} \leq \mathcal{L}_\lambda^{LP} + \lambda$$

In particular, for small $\lambda$ the optimal scores approximately agree. On the other hand, increasing $\lambda$ strengthens the soft constraints, which means that the theoretical observations from Section 4 become more pertinent with growing $\lambda$. Our experiments show exactly the behavior that WGAN-LP and WGAN-GP perform very similarly for small $\lambda$, while WGAN-LP performs much better for larger values of $\lambda$ and its performance is much less dependent on the choice of hyperparameter $\lambda$.

---

[4]While the authors note that "In practice, we found this [using the GP or two-sided penalty] to converge slightly faster and to better optima. Empirically this seems not to constrain the critic too much...", our experiments point towards another conclusion.

**A more general view.** The Kantorovich duality theorem holds in a quite general setting. For example, a different metric can be substituted for the Euclidean distance $||\cdot||_2$. Taking $||\cdot||_2^p$ for a different natural number $p$ for example leads to the minimization of the Wasserstein distance of order $p$ (i.e., the *Wasserstein-p* distance). Based on the dual problem to the computation of the Wasserstein distance of order $p$ (as given by the Kantorovich duality theorem) we still need to maximize Equation (5) with the only difference that 1-Lipschitz-continuity is now measured with respect to $||\cdot||_2^p$. For our training method this entails that the only modification to make is to use the regularization term given by (7), where the Euclidean distance is replaced by the metric of interest. We provide experimental results for the Wasserstein-2 distance in Appendix D.5.

Recently, by Bellemare et al. (2017), the Wasserstein distance was replaced by the energy distance [5]. For the training of Cramer GANs, the authors apply the GP-penalty term proposed by Gulrajani et al. (2017). We expect that using the LP-penalty term instead is also beneficial for Cramer GANs.

## 6 EXPERIMENTS

We perform several experiments on three toy data sets, *8Gaussians*, *25Gaussians*, and *Swiss Roll* [6], to compare the effect of different regularization terms. More specifically, we compare the performance of WGAN-GP and WGAN-LP as described in Equations (6) and (8) respectively, where the penalty was applied to points randomly sampled on the line between the training sample $x$ and the generated sample $y$. Other sampling methods are discussed in Appendix D.4.

Both, the generator network and the critic network, are simple feed-forward NNs with three hidden Leaky ReLU layers, each containing 512 neurons, and one linear output layer. The dimensionality of the latent variables of the generator network was set to two. During training, 10 critic updates are performed for every generator update, except for the first 25 generator updates, where the critic is updated 100 times for each generator update in order to get closer to the optimal critic in the beginning of training. Both networks were trained using RMSprop (Tijmen & Hinton, 2012) with learning rate $5 \cdot 10^{-5}$ and a batch size of 256.

To see whether our findings on toy data sets can be transferred to real world settings, we trained bigger WGAN-GPs and WGAN-LPs on CIFAR-10 as it is described below. Code for the reproduction of our results is available under `https://github.com/lukovnikov/improved_wgan_training`.

**Level sets of the critic.** A qualitative way to evaluate the learned critic function for a two-dimensional data set is by displaying its level sets, as it was done by Gulrajani et al. (2017) and Kodali et al. (2017). The level sets after 10, 50, 100 and 1000 training iterations of a WGAN trained with the GP and LP penalty on the Swiss Roll data set are shown in Figure 3. Similar experimental results for the 8Gaussians and 25Gaussian data sets can be found in Appendix D.1.

It becomes clear that with a penalty weight of $\lambda = 10$, which corresponds to the hyperparameter value suggested by Gulrajani et al. (2017), the WGAN-GP does neither learn a good critic function nor a good model of the data generating distribution. With a smaller regularization parameter, $\lambda = 1$, learning is stabilized. However, with the LP-penalty a good critic is learned even with a high penalty weight in only a few iterations and the level sets show higher regularity. Training a WGAN-LP with lower penalty weight led to equivalent observations (results not shown). We also experimented with much higher values for $\lambda$, which led to almost the same results as for $\lambda = 10$, which emphasizes that LP-penalty based training is less sensitive to the choice of $\lambda$.

**Evolution of the critic loss.** To yield a fair comparison of methods applying different regularization terms, we display values of the critic's loss functions without the regularization term throughout training. Results for WGAN-GPs and WGAN-LPs are shown in Figure 4.

The optimization of the critic with the GP-penalty and $\lambda = 5$ is very unstable: the loss is oscillating heavily around 0. When we use the LP-penalty instead, the critic's loss smoothly reduces to zero,

---

[5]The energy distance (Székely & Rizzo, 2013) has the same convergence properties as the Wasserstein distance, but additionally satisfies that the sample based gradient approximation does not have a bias.

[6]The same data sets were also used in the analysis of Gulrajani et al. (2017).

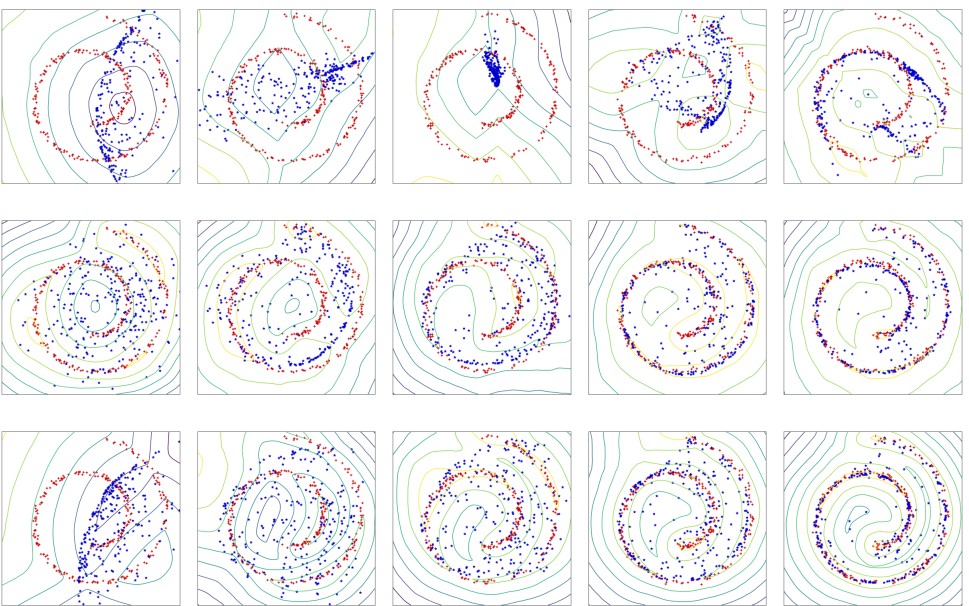

Figure 3: Level sets of the critic $f$ of WGANs during training, after 10, 50, 100, 500, and 1000 iterations. Yellow corresponds to high, purple to low values of $f$. Training samples are indicated in red, generated samples in blue. Top: GP-penalty with $\lambda = 10$. Middle: GP-penalty with $\lambda = 1$. Bottom: LP-penalty with $\lambda = 10$.

which is what we expect when the generative distribution $\nu$ steadily converges to the empirical distribution $\mu$. Also note that we would expect the negative of the critic's loss to be slightly positive, as a good critic function assigns higher values to real data points $x \sim \mu$ and lower values to generated points $y \sim \nu$. This is exactly what we observe when using the LP-penalty Interestingly, when using the LP-penalty in combination with a very high penalty weight, like $\lambda = 100$, we obtain the same results, indicating that the constraint is always fulfilled for $\lambda = 10$ already. Using $\lambda = 1$ in combination with the GP-penalty on the other hand stabilized training but still results in fluctuations in the beginning of the training (results shown in Appendix D.2).

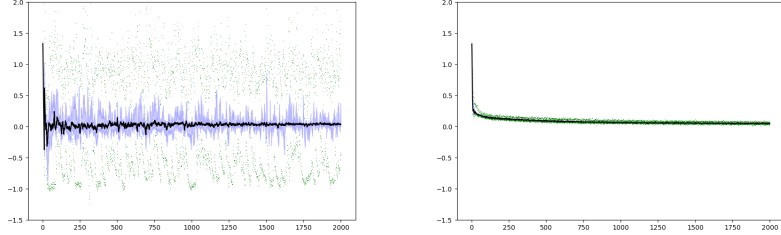

Figure 4: Evolution of the negative of WGAN critic's loss (without the regularization term) for $\lambda = 5$. Median results over the 20 runs (blue area indicates quantiles, green dots outliers). Left: For the GP-penalty. Right: For the LP-penalty.

**Estimating the Wasserstein distance.** In order to estimate how the actual Wasserstein distance between the real and generated distribution evolves during training, we compute the cost of minimum assignment based on Euclidean distance between sets of samples from the real and generated distributions, using the Kuhn-Munkres algorithm (Kuhn, 1955). We use a sample set size of 500 to maintain reasonable computation time and estimate the distance every 10th iteration over the course of 500 iterations. All experiments were repeated 10 times for different random seeds. From the re-

| PENALTY WEIGHT | WGAN-GP | WGAN-LP |
|:---:|:---:|:---:|
| 0.1 | 7.781 ($\pm$ 0.104) | 8.017($\pm$ 0.075) |
| 5 | 7.817 ($\pm$ 0.095) | 7.859 ($\pm$ 0.085) |
| 10 | 7.840 ($\pm$ 0.066) | 7.989 ($\pm$ 0.119) |
| 100 | 7.548 ($\pm$ 0.102) | 7.815 ($\pm$ 0.038) |
| 200 | 7.472 ($\pm$ 0.070) | 7.721 ($\pm$ 0.105) |

Table 1: Inception Score on CIFAR-10. Reported are the maximal mean values reached during training. Means are computed over 10 image sets, variances given in parenthesis.

sults for WGAN-GP and WGAN-LP with $\lambda = 5$ shown in Figure 5, we conclude that the proposed LP-penalty leads to smaller estimated Wasserstein distance and less fluctuations during training.

When training WGAN-GPs with a regularization parameter of $\lambda = 1$, training is stabilized as well (see Appendix D.3), indicating that the effect of using a GP-penalty is highly dependent on the right choice of $\lambda$.

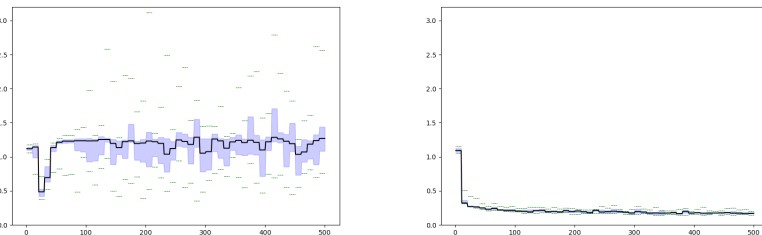

Figure 5: Evolution of the approximated Wasserstein-1 distance during training of WGANs ($\lambda = 5$, median results over 10 runs). Left: For the GP-penalty. Right: For the LP-penalty.

**Sample quality on CIFAR-10.** We trained WGANs with the same ResNet generator and discriminator and the same hyperparameters as Gulrajani et al. (2017) and computed the Inception score (Salimans et al., 2016) throughout training (plots can be found in Appendix D.6). The maximal scores reached in 100000 training iterations with different regularization parameters are reported in Table 1. WGAN-LP reaches the similar or slightly better Inception score as WGAN-GP with small penalty weight ($\lambda \leq 10$), while being more stable to other choices of this hyperparameter. This is especially interesting in the light of a recent large scale study, which also reported a strong dependence of sample quality on $\lambda$ for WGAN-GP (see, Figure 8 and 9 in Lucic et al., 2017). Another interesting observation can be made by monitoring the value of the regularization term during training, as in Figure 6), where contributions to the penalty from $||\nabla f(\hat{x})|| > 1$ are shown in the upper and contributions $||\nabla f(\hat{x})|| < 1$ (only existing for WGAN-GP) are shown in the lower half plane. While the values of the one-sided regularization of WGAN-LP are only slightly larger for larger $\lambda$ (100 compared to 5) the regularization of WGAN-GP shows a strong dependence on the choice of the regularization parameter. For $\lambda = 5$ the penalty contributions from gradient norms smaller than one almost vanished (we found this getting even more severe for even smaller regularization parameters). That is, in a setting where WGAN-GP is performing fine it actually acts similar to WGAN-LP.

**Related penalties** We tested the effects of using the regularization terms given by Equation (7) and Equation (9) instead of the the proposed regularization given in Equation (8). Both lead to good performance on toy data but to considerably worse results on CIFAR-10, where training was very unstable. Results are shown in Appendix D.7

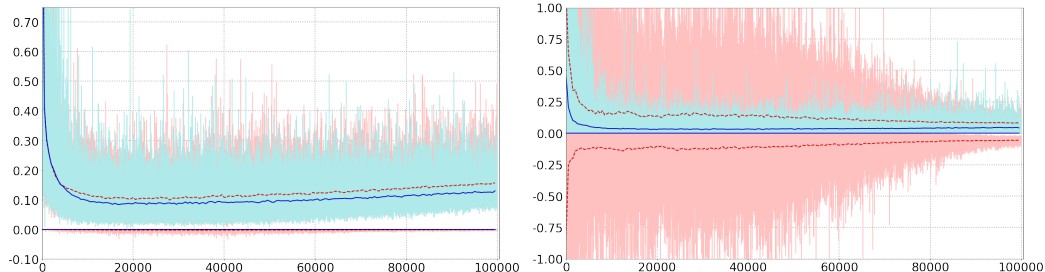

Figure 6: Comparison of the magnitude of the gradient penalty during training on CIFAR, showing $< 1$ and $> 1$ contributions (i.e. $\min(0, ||\nabla f(\hat{x})|| - 1)^2$ resp. $\max(0, ||\nabla f(\hat{x})|| - 1)^2$). Left: for regularization parameter $\lambda = 5$. Right: for regularization parameter $\lambda = 100$. The (one-sided) gradient penalty of WGAN-LP is depicted in blue (solid), the gradient penalty of WGAN-GP in red (dashed). All the values for every iteration (one mini-batch) are shown in light blue and red. Dark blue and red lines show the mean over a sliding window of size 500. The figure shows that the part of the gradient penalty of WGAN-GP penalizing a gradient $\leq 1$ almost vanishes for a small regularization parameter, bringing it close to WGAN-LP. For larger values of the regularization parameter, the total penalty of WGAN-GP and its contributing parts are larger than the penalty of WGAN-LP, however, the performance of WGAN-GP suffers more.

## 7 CONCLUSION

For stable training of Wasserstein GANs, we propose to use the following penalty term to enforce the Lipschitz constraint that appears in the objective function:

$$\mathbb{E}_{\hat{x} \sim \tau} [(\max\{0, ||\nabla f(\hat{x})|| - 1\})^2] \ .$$

We presented theoretical and empirical evidence that this gradient penalty performs better than the previously considered approaches of clipping weights and of applying the stronger gradient penalty given by $\mathbb{E}_{\hat{x} \sim \tau}[(||\nabla f(\hat{x})||_2 - 1)^2]$. In addition to more stable learning behavior, the proposed regularization term leads to lower sensitivity to the value of the penalty weight $\lambda$ (demonstrating smooth convergence and well-behaved critic scores throughout the whole training process for different values of $\lambda$).

### ACKNOWLEDGMENTS

This work is supported in part by the European Union under the Horizon 2020 Framework Program for the project WDAqua (GA 642795).

The authors thank the anonymous reviewers for their valuable suggestions.

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

## A    PROPERTIES OF AN OPTIMAL CRITIC FUNCTION OF WGANS

An issue of the original GAN discriminator was that it outputs zero every time it is certain to see generated data, independent on how far away a generated data point lies from the real distribution. As a consequence, locally, there is no incentive for the generator to rather generate a value closer to (but still off) the real data; GAN critic's optimal value is zero in either case. The WGAN's optimal critic function measures this distance which helps for the generated distribution to converge, but the interpretation of the absolute value as indicating real (close to 1) and fake data (close to 0) is lost. And worse, there is even no guarantee that the relative values of the optimal critic function help to decide what is real and what is fake. Although this does not seem to cause major problems for the iterative training procedure in practice, we still consider it worthwhile to give a specific example justifying the following observation.

**Observation 4.** *The WGAN generator could learn wrong things, basing its decision on the values of the optimal critic function, i.e., if it generates at locations of high critic function values.*

Consider the following setting, where the X's represent generated and the O's represent real data points.

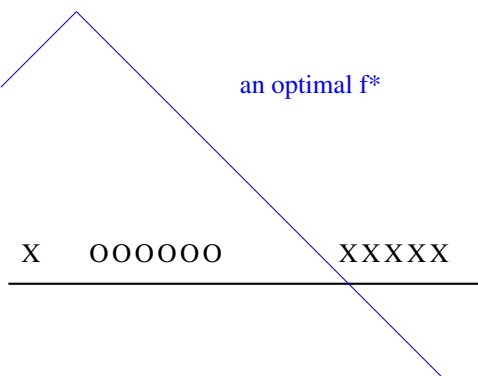

Figure 7: Values of the WGAN critic function for some generated data points can be higher than the critic's values for some real data points. Thus, fake and real points can not be distinguished based on the critics values alone. Real data points are represented by O, generated by X.

An optimal coupling in this example is quite obvious: We connect the left-most O with the X on the left, and then extend by an arbitrary matching of the other O's with the other X's. It is then not hard to verify that the indicated critic function with slope $1$ or $-1$ almost everywhere leads to an equality in the Kantorovich duality and hence is optimal. The value of the critic function at the left-most X is higher than the value at the right-most O, suggesting to generate images at the wrong position. This issue might be fixed by the alternating updates of generator and critic at a later stage of training when less X's are generated so far on the right side of the O's. The critic function will then flatten the peak, eventually assigning a lower value to an X on the left than to any of the O's.

**Remark 1.** *The same holds (with only a slight change of the critic function $f$) if the $X$'s and $O$'s denote the centers of Gaussians. This can be shown with similar arguments as those in the proofs in Appendix C.3.*

### A.1    PROVING OPTIMALITY OF CERTAIN COMBINATIONS OF COUPLING AND CRITIC FUNCTION

We show here that the coupling and critic function indicated in Figure 1 are indeed optimal.

In the one-dimensional example on the left, $\int_{\mathbb{R}\times\mathbb{R}} |x-y|\, d\pi^*(x,y) = \frac{1}{7}(1+1+1+1+1+1+1) = \int_{\mathbb{R}} f^*(x)\, d\mu(x) - \int_{\mathbb{R}} f^*(x)\, d\nu(x)$ and thus $\pi^*$ and $f^*$ are indeed optimal.

In the two-dimensional example on the right, the coupling indicated in red and the critic function (described by its function values in blue) are optimal, since with this choice we have

$\int_{\mathbb{R}^n \times \mathbb{R}^n} ||x - y||_2 \ d\pi^*(x, y) = \frac{1}{2}(1 + 1) = 1$ and $\int_{\mathbb{R}^n} f^*(y) \ d\nu(y) - \int_{\mathbb{R}^n} f^*(x) \ d\mu(x) = \frac{1}{2}(1 + a + 1) - \frac{1}{2}(0 + a) = 1$. Equality of the left hand side and right hand side of the equation proves optimality on both sides.

## B   THE ISSUE WITH THE WEIGHT CLIPPING APPROACH

The critic function of WGAN is given by a neural network, which raises the question on how to enforce the 1-Lipschitz constraint in the maximization problem of the objective in Equation (5). As Arjovsky et al. (2017) point out, it does not matter whether to maximize over 1-Lipschitz or $\alpha$-Lipschitz continuous functions, since we can equivalently optimize $\alpha \cdot W(\mu, \nu)$ instead of $W(\mu, \nu)$. An easy consideration leads to the following lemma.

**Lemma 1.** *The optimal critic function $f^*$ (leading to the maximum in Eq. (4)) exhausts the Lipschitz constraint for given $\alpha$ in the sense that there is a pair of points $(x, y)$ such that $f^*(x) - f^*(y) = \alpha||x - y||_2$.*

*Proof.* If $\sup_{x \neq y} \left\{ \frac{f^*(y) - f^*(x)}{||x - y||_2} \right\} = c < \alpha$ , then $g = \frac{1}{c} f^*$ generates a contradiction to the optimality of $f^*$. (Alternatively, in the case $\alpha = 1$, it follows directly from Theorem 1, (ii), that the transport is optimal if and only if the Lipschitz constraint of one is exhausted for any two points of the coupling.) □

**Observation 5.** *Weight clipping is not a good strategy to enforce the Lipschitz constraint for the critic function.*

First note that by clipping the weights we enforce a common Lipschitz constraint, where the common Lipschitz constant $\bar{\alpha}$ is defined as the minimal $\alpha \in \mathbb{R}$ such that $f(x) - f(y) \leq \alpha||x - y||_2$ for all $x, y$ and all functions $f$ that can be generated by the network under weight clipping. The actual value of $\bar{\alpha}$ does not follow directly from the weight clipping constant $c_{\max}$ but can be computed from the structure of the network. From Lemma 1 we know that an optimal $f^*$ exhausts the Lipschitz constraint. We will now show exemplarily for deep NN with rectified linear unit (ReLU) activation functions that there is an extremely limited number of functions generated by the NN using weight clipping that do exhaust the implicitly given common Lipschitz constraint $\bar{\alpha}$. It follows that, in almost all cases, the optimal $f^*$ is not in the class of functions that can be generated by the network under the weight clipping constraint.

**Proposition 4.** *Consider a (deep) NN with ReLU activation functions and linear output layer. A function generated by the NN under constraining each weight in absolute value by $c_{max}$ exhausts the common Lipschitz constraint if and only if*

(a) *The weight matrix of the first layer consists of constant columns with value $c_{max}$ or $-c_{max}$.*

(b) *The weights of all other layers are given by a matrix $C^{max}$ with every entry equal to $c_{max}$.*

*Proof.* We need to determine every function $f^*$ generated by the neural network, such that we can find points $x^* \neq y^*$ with $f^*(y^*) - f^*(x^*) = \bar{\alpha}||x^* - y^*||_2$. Recall that $\bar{\alpha}$ is defined as the minimal $\alpha$ satisfying $f(y) - f(x) \leq \alpha||x - y||_2$ for all functions $f$ generated by the neural network and all points $x, y$.

In the following, we will denote by $\alpha(f)$ the Lipschitz constant of $f$, i.e., the smallest $\alpha \in \mathbb{R}$ such that $f(x) - f(y) \leq \alpha||x - y||_2$ for all $x, y$.

Every function generated by the neural net is a composition of functions

$$f = f_n \circ \text{relu} \circ f_{n-1} \circ \ldots \circ \text{relu} \circ f_1.$$

with linear functions $f_i$ and relu denoting a layer of activation functions with rectifier linear units. Since each linear function $f_i$ is Lipschitz continuous with Lipschitz constant $\alpha(f_i)$ and relu is Lipschitz continuous with $\alpha(\text{relu}) = 1$, it follows that $f$ is Lipschitz continuous with $\alpha(f) \leq \prod_i \alpha(f_i)$. Moreover, equality holds if there is a pair of points $(x, y)$ such that the consecutive images witness the maximal Lipschitz constant $\alpha(f_i)$ and $\alpha(\text{relu})$ for each of the individual functions making up the composition of $f$. More formally, equality holds if and only if there is a tuple of pairs of points $(x^{(i)}, y^{(i)}), 1 \leq i \leq n - 1$, such that for all $1 \leq i \leq n$,

(i) $x^{(i)} \neq y^{(i)}$

(ii) $x^{(i+1)} = \text{relu} \circ f_i(x^{(i)})$ and $y^{(i+1)} = \text{relu} \circ f_i(y^{(i)})$

(iii) $|f_i(x^{(i)}) - f_i(y^{(i)})| = \alpha(f_i)||x^{(i)} - y^{(i)}||_2$

(iv) All entries of $f_i(x^{(i)})$ and $f_i(y^{(i)})$ are larger or equal to zero. This is equivalent to the condition that

$$|\text{relu}(f_i(x^{(i)})) - \text{relu}(f_i(y^{(i)}))| = \alpha(\text{relu})||f_i(x^{(i)}) - f_i(y^{(i)})||_2 \ .$$

It follows that to determine $f^*$ we need to maximize $\alpha(f_i)$ for the linear layers with weight constraint $c_{\max}$ and find a sequence of points $(x^{(i)}, y^{(i)})$ that satisfy (i)-(iv). The existence of the sequence of points shows that $\alpha(f^*) = \prod_{i=1}^n \alpha(f_i)$ and maximizing each $\alpha(f_i)$ then shows that

$$\alpha(f^*) = \prod_{i=1}^n \alpha(f_i) = \bar{\alpha} \ .$$

Since, as we will show, the conditions in (a) and (b) maximize the Lipschitz constraint of each layer individually, the existence of suitable $(x^{(i)}, y^{(i)})$ proves the if-direction of the proposition.

For the only-if direction, we will see that the ability to find the sequence of points gives restrictions on how to maximize $\alpha(f_i)$ of an individual layer, leading to the more restrictive condition of (b) for all but the first layer (cf. (a)).

So let us first maximize the Lipschitz constraint of each linear layer and then make sure that we can find the corresponding points. We write the linear layer as a matrix multiplication $f_i(x) = A^{(i)}x$. Using linearity,

$$\alpha(f_i) = \max_{||z||_2=1} ||A^{(i)}z||_2 \ ,$$

and our goal can be reformulated to finding the matrix $A^{(i)}$ maximizing $\alpha(f_i)$.

For any fixed $z$, $||A^{(i)}z||_2$ is maximized exactly when each vector entry is maximized in absolute value. Now, with $A^{(i)} = (a_{j,k}^{(i)})_{j,k}$ and $\text{sgn}(\cdot)$ denoting the sign function,

$$|(A^{(i)}z)_j| = \left|\sum_k a_{j,k}^{(i)} z_k\right| \leq \sum_k |a_{j,k}^{(i)}||z_k| \leq \sum_k c_{\max}|z_k| \text{ (by the weight constraint)}$$

$$= \sum_k (c_{\max} \cdot \text{sgn}(z_k)) \cdot z_k$$

and equality holds if and only if $A^{(i)}$ or $-A^{(i)}$ consists of columns of constant entry with the value $c_{\max} \cdot \text{sgn}(z_k)$ in column $k$. It follows that

$$\alpha(f_i) = \max_{||z||_2=1} ||A^{(i)}z||_2 = \max_{||z||_2=1} c_{\max} \cdot ||z||_1$$

$$\leq \max_{||z||_2=1} c_{\max}\sqrt{dim(z)} \cdot ||z||_2 \text{ (by Cauchy-Schwartz inequality)}$$

$$= c_{\max}\sqrt{dim(z)}$$

with equality if and only if $z_k = \pm\dfrac{1}{\sqrt{dim(z)}}$ for all $k$.

Hence, for the first linear layer we need to choose a matrix $A^{(1)}$ satisfying (a) of the statement of the proposition.

Now, we find a pair $(x^{(1)}, y^{(1)})$ with

$$x^{(1)} - y^{(1)} = a \cdot (\pm 1, \pm 1, \ldots, \pm 1) \text{ for some } a \neq 0$$

such that

$$\text{sgn}(x_k^{(1)}) = \text{sgn}(y_k^{(1)}) = \text{sgn}(x_k^{(1)} - y_k^{(1)}) = \text{the sign of column } k \text{ of } A^{(1)}.$$

This is the only possibility to ensure (iii) and (iv) of the conditions above. Note that also (i) holds for $(x^{(1)}, y^{(1)})$, and (ii) (together with (iv)) determines $(x^{(2)}, y^{(2)})$ uniquely from $(x^{(1)}, y^{(1)})$ as

$$
\begin{aligned}
x^{(2)} &= A^{(1)} x^{(1)} = c_{\max} \cdot ||x^{(1)}||_1 \cdot (1, 1, ..., 1) \\
\& \quad y^{(2)} &= A^{(1)} y^{(1)} = c_{\max} \cdot ||y^{(1)}||_1 \cdot (1, 1, ..., 1)
\end{aligned}
$$

We may assume that $||x^{(1)}||_1 > ||y^{(1)}||_1$. (Otherwise, switch the roles of $x$ and $y$. In the case of equality, we need to choose a different pair for $(x^{(1)}, y^{(1)})$ not to violate (i) for $(x^{(2)}, y^{(2)})$.) Then we have that $x^{(2)} \neq y^{(2)}$,

$$+1 = \operatorname{sgn}(x_k^{(2)}) = \operatorname{sgn}(y_k^{(2)}) = \operatorname{sgn}(x_k^{(2)} - y_k^{(2)}) \text{ for all } k.$$

Using the same arguments as above, it follows that for such $(x^{(2)}, y^{(2)})$, to maximize the Lipschitz constant of $f_2$ (and to guarantee that the maximum is reached at $(x^{(2)}, y^{(2)})$), we need to have $A^{(2)}$ equal to a matrix with $c_{\max}$ at each position.

Now (i)-(iv) also hold for the second layer and one may now proceed by induction to show that for $i \geq 2$, $A^{(i)}$ contains only $c_{\max}$ for each of its entries. This is the only way to maximize the Lipschitz constraint for functions generated by the neural net, and it does indeed hold $||f^*(x^*) - f^*(y^*)||_2 = \bar{\alpha}||x - y||_2$ with $x^* = x^{(1)}, y^* = y^{(1)}$ and

$$(x^{(i)}, y^{(i)}) = (f_i \circ \operatorname{relu} \circ f_{i-1} \circ \ldots \circ \operatorname{relu} \circ f_1(x^*), f_i \circ \operatorname{relu} \circ f_{i-1} \circ \ldots \circ \operatorname{relu} \circ f_1(y^*)).$$

$\square$

## C  PROOFS

### C.1  PROOF OF THEOREM 1

*Proof.* We provide the arguments how to derive our version from Theorem 5.10 of Villani (2008).

With $c(x, y) = ||x - y||_2$, our assumptions imply (with $c_{\mathcal{X}} = c_{\mathcal{Y}} = ||\cdot||_2$) that all conclusions of Theorem 5.10 $(i) - (iii)$ hold. Moreover, 5.4 of Villani (2008) shows that in this case $\psi = \psi^c$ (in the notation of Villani (2008)) and $c$-convexity is the same as 1-Lipschitz continuity. This leads to our formulation in (i) and the existence of an optimal coupling $\pi^*$ and an optimal critic function $f^*$ by part (iii).

If we let

$$\Gamma_f = \{(x, y) \in \mathbb{R}^n \times \mathbb{R}^n \mid f(x) - f(y) = ||x - y||_2\}$$

then it follows from the proof of Theorem 5.10 that the set $\Gamma$ in part 5.10 (iii) is given by $\Gamma = \bigcap_{f^* \in \mathcal{L}ip_1 \text{ optimal}} \Gamma_{f^*}$, where $f^*$ being optimal means that it leads to a maximum on the RHS of equation (1).

To prove our part $(ii)$ from 5.10, let $\pi^*$ be optimal. Then, by 5.10 (iii), $\pi^*(\Gamma) = 1$. Hence, in particular, $\pi^*(\Gamma_{f^*}) = 1$ for all optimal $f^* \in \mathcal{L}ip_1$. This shows that (a) implies (b). For the other direction, we use that if $\pi^*(\Gamma_{f^*}) = 1$ for all optimal $f^*$, then $\pi^*(\Gamma) = 1$, which by Theorem 5.10 (iii) is equivalent to $\pi^*$ being optimal. $\square$

### C.2  PROOF OF PROPOSITION 1

*Proof.* It follows from Theorem 1 (ii) that for all $(x, y)$ in the support of $\pi^*$ we have $|f^*(y) - f^*(x)| = ||x - y||_2$. Considering the line between $x$ and $y$, the 1-Lipschitz constraint implies that the values of $f^*$ have to follow a linear function (since assuming that the slope was smaller than one at some point would imply that the differentiable function must have a slope larger than one somewhere else between $x$ and $y$, which contradicts the 1-Lipschitz constraint). It follows that at each point on the line, the partial derivative has norm equal to one into the direction pointing from the real data point $x$ to the generated one $y$ (which are coupled by the corresponding optimal coupling). Since, by the 1-Lipschitz constraint, the maximal norm of a partial derivative at any point into any direction is one, the given direction is the direction of maximal descent, i.e. equals the gradient. $\square$

### C.3 Proof of Proposition 2

To prove Proposition 2, we first prove that $\phi^*(x) = -|x|$ is the optimal critic function for certain distributions with non-overlapping support, and then reduce the example with Gaussian functions to this simplified setting.

**Proposition 5.** *Let $f$ and $g$ be two continuous functions on the real line that satisfy the following conditions:*

- *$f$ and $g$ are symmetric with respect to the y-axis.*

- *$f(x) \geq 0$ and $g(x) \geq 0$ for all $x$.*

- *If $supp_\circ(h) = \{x \in \mathbb{R} \mid h(x) > 0\}$ denotes the open support of a continuous function $h$, then $supp_\circ(f) \cap supp_\circ(g) = \emptyset$.*

- *$f$ has connected support (this implies that $f$ is centered around $0$ because of the symmetry).*

- *$\int_\mathbb{R} f(x)dx = \int_\mathbb{R} g(x)dx$.*

*Then the maximum of $\int_\mathbb{R} \phi(x)(f(x) - g(x))dx$ over $\phi \in \mathcal{L}ip_1$ is maximized for $\phi^*(x) = -|x|$.*

*Proof.* Before going into the technical details, we wish to point out the simple idea of the proof, which is to transport the left/right half of the distribution given by $g$ to the left/right half of the distribution given by $f$ respectively.

We first multiply both $f$ and $g$ by a constant number $c$ such that

$$\int_\mathbb{R} c \cdot f(x)dx = \int_\mathbb{R} c \cdot g(x)dx = 1.$$

Then $c \cdot f$ and $c \cdot g$ define probability density functions. A function $\phi \in \mathcal{L}ip_1$ maximizes $\int_\mathbb{R} \phi(x)(c \cdot f(x) - c \cdot g(x))dx$ if and only if it maximizes $\int_\mathbb{R} \phi(x)(f(x) - g(x))dx$. We therefore may assume from now on that

$$\int_\mathbb{R} f(x)dx = \int_\mathbb{R} g(x)dx = 1.$$

Now it suffices to find a coupling $\pi$ of the probability distributions defined by $f$ and $g$ (that is itself defined by a probability density function $\pi : \mathbb{R} \times \mathbb{R} \to \mathbb{R}$) such that for $\phi(x) = -|x|$ we get

$$\int_{\mathbb{R} \times \mathbb{R}} |x - y| \cdot \pi(x, y)dxdy = \int_\mathbb{R} \phi(x)(f(x) - g(x))dx.$$

The proof then follows from the Kantorovich duality theorem 1, because the right hand side is always smaller or equal to the left hand side for arbitrary coupling $\pi$ and function $\phi \in \mathcal{L}ip_1$ and is consequently maximized when equality holds. By the assumption of symmetry, we may write $g = g_1 + g_2$ where the support $supp(g_1) \subseteq \{x \mid x < 0\}$ and $g_2(x) = g_1(-x)$ for all $x$. The area under $g_1(x)$ equals half the area under $f(x)$, or put differently,

$$\int_\mathbb{R} g_1(x)dx = \int_\mathbb{R} f(x)\delta_{(-\infty,0]}(x)dx = \frac{1}{2}.$$

We now consider the probability density function $\pi_1 : \mathbb{R} \times \mathbb{R} \to \mathbb{R}$ given by

$$\pi_1(x, y) = 2g_1(x) \cdot 2f(y) \cdot \delta_{(-\infty,0]}(y),$$

which defines a coupling between the two distributions given by the probability density functions $2g_1$ and $2f \cdot \delta_{(-\infty,0]}$. For later use we note that

$$\int_{x \in \mathbb{R}} \pi_1(x, y) \, dx = 2 \cdot f(y) \cdot \delta_{(-\infty,0]}(y) \text{ and } \int_{y \in \mathbb{R}} \pi_1(x, y) \, dy = 2 \cdot g_1(x).$$

We define $\pi_2(x, y) = \pi_1(-x, -y)$ for $y \neq 0$ and $\pi_2(x, 0) = 0$. Further, we let $\pi = \frac{1}{2}\pi_1 + \frac{1}{2}\pi_2$. Then $\pi$ defines a coupling between $g$ and $f$ as can be seen by computing

$$\int_{x \in \mathbb{R}} \pi(x, y) dx = \frac{1}{2} \int_{x \in \mathbb{R}} \pi_1(x, y) dx + \frac{1}{2} \int_{x \in \mathbb{R}} \pi_2(x, y) dx$$

$$= \frac{1}{2} \int_{x \in \mathbb{R}} \pi_1(x, y) dx + \frac{1}{2} \int_{x \in \mathbb{R}} \pi_1(-x, -y) \delta_{\{y \neq 0\}}(y) dx$$

$$= f(y) \delta_{(-\infty, 0]}(y) + f(y) \delta_{(0, \infty)}(y) = f(y)$$

and

$$\int_{y \in \mathbb{R}} \pi(x, y) dy = \frac{1}{2} \int_{y \in \mathbb{R}} \pi_1(x, y) dy + \frac{1}{2} \int_{y \in \mathbb{R}} \pi_2(x, y) dy$$

$$\frac{1}{2} \int_{y \in \mathbb{R}} \pi_1(x, y) dy + \frac{1}{2} \int_{y \in \mathbb{R}} \pi_1(-x, -y) \delta_{\{y \neq 0\}}(y) dy$$

$$= g_1(x) + g_1(-x) = g_1(x) + g_2(x) = g(x)$$

We have established the existence of some coupling between $f$ and $g$ and we will now compute its transport costs. We will subsequently show that this equals $\int_{\mathbb{R}} (-|x|)(f(x) - g(x)) dx$, hence both $\pi$ and $\phi$ are optimal by realizing the Kantorovich duality.

We aim at showing $\int_{\mathbb{R} \times \mathbb{R}} |x - y| \pi(x, y) dx dy = \int_{\mathbb{R}} (-|x|)(f(x) - g(x)) dx$.

$$\int_{\mathbb{R} \times \mathbb{R}} |x - y| \pi(x, y) dx dy \overset{symmetry}{=} \int_{\mathbb{R} \times \mathbb{R}} |x - y| \pi_1(x, y) dx dy$$

$$= \int_{\mathbb{R} \times \mathbb{R}} (y - x) \pi_1(x, y) dx dy.$$

The latter equation holds because for $(x, y)$ in the support of $\pi_1$ we have $x \leq y$. (To see this, note that support of $\pi_1$ is a subset of the support of $g_1 \times (f \cdot \delta_{(-\infty, 0]})$.) Let

$$x_0 = \frac{\int_{\mathbb{R}} x \cdot g_1(x) dx}{\int_{\mathbb{R}} g_1(x) dx}, \text{ and } y_0 = \frac{\int_{\mathbb{R}} y \cdot f(y) \cdot \delta_{(-\infty, 0)}(y) dy}{\int_{\mathbb{R}} f(y) \cdot \delta_{(-\infty, 0)}(y) dy}.$$

Then

$$\int_{\mathbb{R}} (x - x_0) \cdot g_1(x) dx = 0 \text{ and } \int_{\mathbb{R}} (y - y_0) \cdot f(y) \cdot \delta_{(-\infty, 0]}(y) dy = 0.$$

Now, it follows that

$$\int_{\mathbb{R} \times \mathbb{R}} (y - x) \pi_1(x, y) dx dy = \int_{\mathbb{R} \times \mathbb{R}} (y - y_0 + y_0 - x) \pi_1(x, y) dx dy$$

$$= \int_x \int_y (y - y_0) \pi_1(x, y) dx dy + \int_x \int_y (y_0 - x) \pi_1(x, y) dx dy$$

$$= \int_y (y - y_0) \int_x \pi_1(x, y) dx dy + \int_x \int_y (y_0 - x) \pi_1(x, y) dx dy$$

$$= 2 \underbrace{\int_y (y - y_0) \cdot f(y) \cdot \delta_{(-\infty, 0]}(y) dy}_{=0} + \int_x \int_y (y_0 - x_0 + x_0 - x) \pi_1(x, y) dx dy$$

$$= (y_0 - x_0) \int_x \int_y \pi_1(x, y) dx dy + \int_x (x_0 - x) \underbrace{\int_y \pi_1(x, y) dy}_{=2g_1(x)} dx$$

$$= (y_0 - x_0).$$

Hence,

$$\int_{\mathbb{R} \times \mathbb{R}} |x - y| \pi(x, y) dx dy = (y_0 - x_0) = \frac{\int_{\mathbb{R}} y \cdot f(y) \cdot \delta_{(-\infty, 0)}(y) dy}{\frac{1}{2}} - \frac{\int_{\mathbb{R}} x \cdot g_1(x) dx}{\frac{1}{2}}.$$

$$= 2 \int_{\mathbb{R}} x \cdot (f(x) \cdot \delta_{(-\infty,0)}(x) - g_1(x)) dx$$

$$= 2 \int_{-\infty}^{0} x \cdot (f(x) - g(x)) dx$$

$$= 2 \int_{-\infty}^{0} (-|x|) \cdot (f(x) - g(x)) dx$$

$$\overset{symmetry}{=} \int_{\mathbb{R}} (-|x|) \cdot (f(x) - g(x)) dx$$

$\square$

We are now able to proof Proposition 2

*Proof to Proposition 2.* Let $f$ denote the probability density function of $\mathcal{N}(0,1)$ and $g = \frac{1}{2}g_{-1} + \frac{1}{2}g_1$ denote the sum of half the probability density functions $g_{-1}$ of $\mathcal{N}(-1,1)$ and $g_1$ of $\mathcal{N}(1,1)$. Let

$$\tilde{f}(x) = \max\{0, (f(x) - g(x))\} \text{ and } \tilde{g}(x) = \max\{0, (g(x) - f(x))\},$$

i.e. $\tilde{f}$ and $\tilde{g}$ are the positive and the negative part of $(f - g)$. Then $\tilde{f}$ and $\tilde{g}$ satisfy the hypothesis of Proposition 5 and the maximum

$$\max_{\phi \in \mathcal{L}ip_1} \int_{\mathbb{R}} \phi(x)(f(x) - g(x)) dx = \max_{\phi \in \mathcal{L}ip_1} \int_{\mathbb{R}} \phi(x)(\tilde{f}(x) - \tilde{g}(x)) dx$$

is obtained for $\phi^*(x) = -|x|$. $\square$

## C.4 PROOF OF PROPOSITION 3

*Proof.* For any fixed function $f$ and $\lambda > 0$, the two regularized losses of the critic function $f$ are of the form

$$\mathcal{L}_\lambda^{LP}(f) = c + \lambda \int \max\{0, (h(z) - 1)\}^2) \mathrm{d}\tau(z) \text{ and } \mathcal{L}_\lambda^{GP}(f) = c + \lambda \int (h(z) - 1)^2) \mathrm{d}\tau(z)$$

for some real number $c$, a function $h$ with with $h(z) \geq 0$ for all $z$ and a probability distribution $\tau$. Since for any real number $0 \leq a$ we have that

$$\max\{0, (a - 1)\}^2 \leq (a - 1)^2 \leq \max\{0, (a - 1)\}^2 + 1$$

it follows that

$$\mathcal{L}_\lambda^{LP}(f) \leq \mathcal{L}_\lambda^{GP}(f) \leq \mathcal{L}_\lambda^{LP}(f) + \lambda.$$

Therefore the inequalities also hold for the infimum over a class of functions, hence

$$\mathcal{L}_\lambda^{LP} \leq \mathcal{L}_\lambda^{GP} \leq \mathcal{L}_\lambda^{LP} + \lambda.$$

$\square$

## C.5 THE ARGUMENTS SUPPORTING OBSERVATION 3

For the coupled pairs $(x, y)$ and $(x, y')$ we have that the partial derivatives at $x$ into the directions of $y$ and $y'$ respectively have an absolute value of one. If there are two such directions and $f^*$ is differentiable, then the norm of its gradient must be larger than one, contradicting the 1-Lipschitz constraint. Indeed, recall that, considering $f$ as a function on the line $\{x + \lambda \cdot v \mid \lambda \in \mathbb{R}\}$ with $v$ of unit length, the slope of $f$ at $x$ is given by $\nabla f(x) \cdot v = D_v(f(x))$. Now

$$\nabla f(x) \cdot v = ||\nabla f(x)||_2 \cdot \cos(\theta_v) \quad (10)$$

with $\theta_v$ being the angle between the vector $\nabla f(x)$ and the unit vector $v$. Equation (10) with $\cos(\theta_v) = 1$ has a unique solution for $v$ with $v = \frac{\nabla f(x)}{||\nabla f(x)||_2}$. It follows that, if for two different directions $v, v'$ we have $D_v(f(x)) = D_{v'}(f(x)) = 1$, then $\cos(\theta_v) = \cos(\theta_{v'}) < 1$ and $||\nabla f(x)||_2 > 1$.

# D ADDITIONAL EXPERIMENTAL RESULTS

## D.1 LEVEL SETS OF THE CRITIC

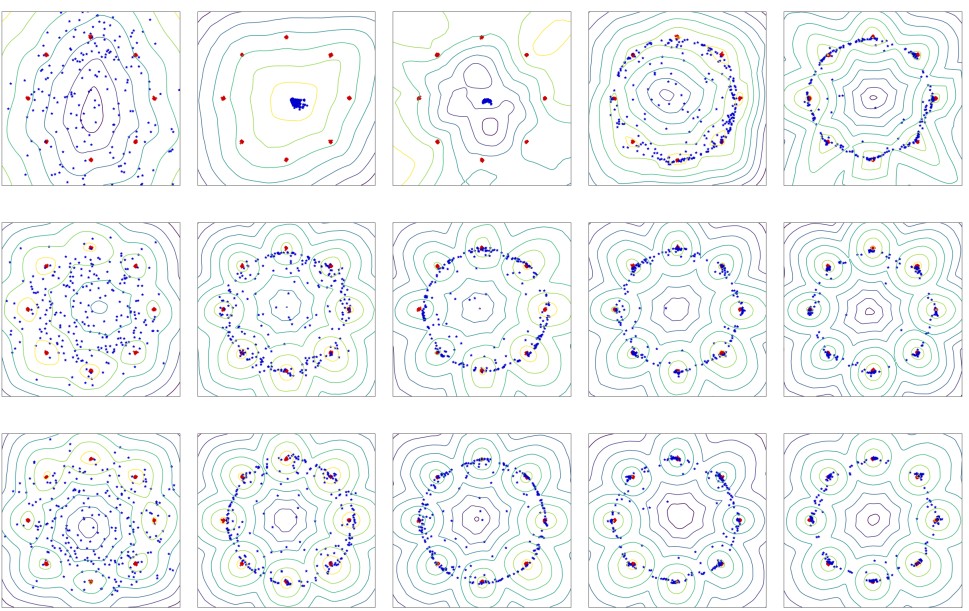

Figure 8: Level sets of the critic (yellow corresponds to high, purple to low values) of WGANs during training (after 10, 50, 100, 500, and 1000 iterations) on the 8Gaussian data set. Top: GP-penalty ($\lambda = 10$). Middle: GP-penalty ($\lambda = 1$). Bottom: LP-penalty ($\lambda = 10$).

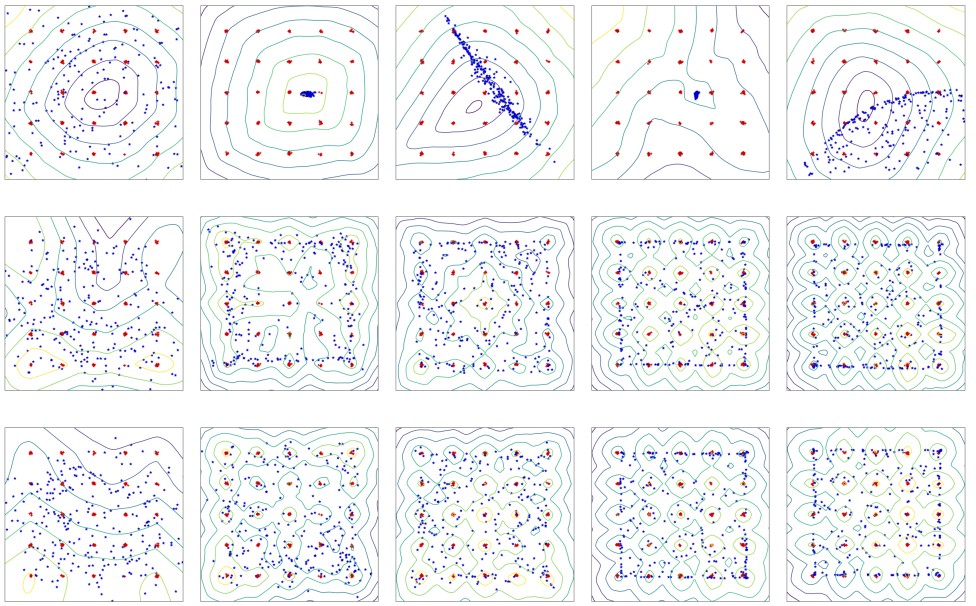

Figure 9: Level sets of the critic (yellow corresponds to high, purple to low values) of WGANs during training (after 10, 50, 100, 500, and 1000 iterations) on the 25Gaussian data set. Top: GP-penalty ($\lambda = 10$). Middle: GP-penalty ($\lambda = 1$). Bottom: LP-penalty ($\lambda = 10$).

## D.2 EVOLUTION OF CRITICS LOSS

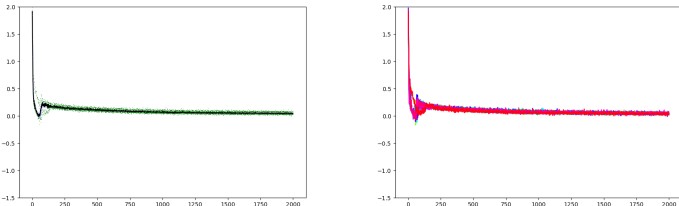

Figure 10: Evolution of the WGAN-GP critics loss without the regularization term ($\lambda = 1$). Left: Median results over the 20 runs (blue area indicates quantiles, green dots outliers). Right: Single runs.

## D.3 EVOLUTION OF THE EM DISTANCE

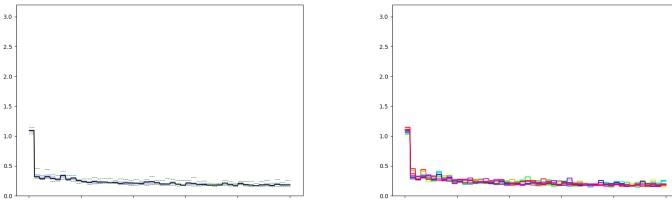

Figure 11: Evolution of the approximated EM distance during training WGAN-GPs with $\lambda = 1$. Left: Median results over the 10 runs. Right: Single runs.

## D.4 DIFFERENT SAMPLING METHODS

We analyzed the effect of the GP- and the LP-penalty using different sampling procedures. In particular, we compared the sampling procedure proposed by Gulrajani et al. (2017) with variants, which generate the samples used for the regularization term by adding random noise either onto training points or onto both training and generated samples. We refer to this as "local perturbation" in the following.[7] The evolution of the critics loss when using this local perturbation can be seen in Figure 12. Results are qualitatively similar to those when using the sampling procedure proposed by Gulrajani et al. (2017). Interestingly, WGAN-GP training is stabilized at a later stage if one only adds noise to training examples and not to generated examples. This indicates that enforcing the GP-penalty close to the data manifold is less harmful. However, the critic's loss is still much more fluctuating than when training a WGAN-LP.

The evolution of the approximated EM distance when using local perturbation (by adding noise to the training examples only) is shown in Figure 13. Training with the GP-penalty leads to larger fluctuations of the approximated Wasserstein-1 distance than training with the LP-penalty. However, fluctuations are less severe compared to the setting when the GP-penalty is used in combination with the sampling procedure proposed by Gulrajani et al. (2017).

---

[7]Note that applying the GP-penalty on samples generated by adding noise to the training examples was also suggested by Kodali et al. (2017), but using a different (asymmetric) noise distribution and in combination with the vanilla GAN objective.

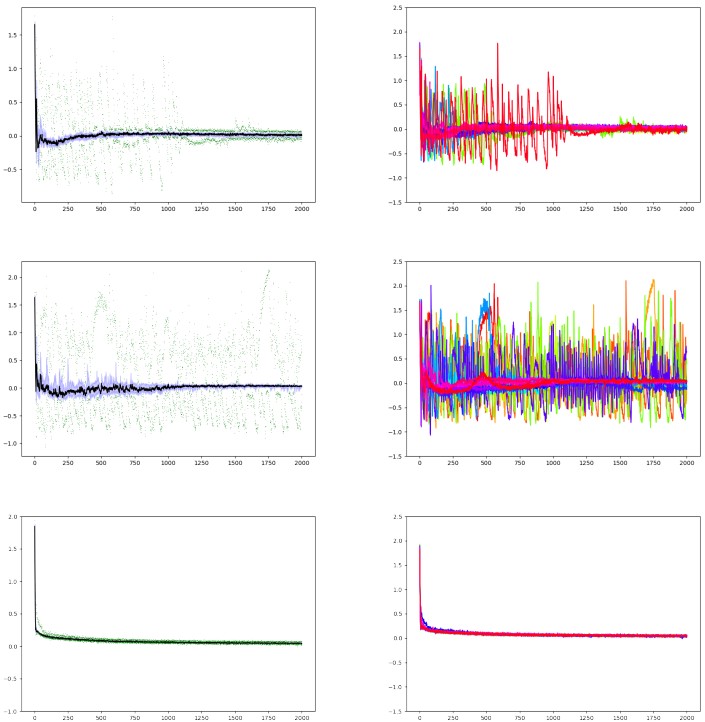

Figure 12: Evolution of the WGAN critic's negative loss with local sampling (without the regularization term). Left: Median results over the 20 runs. Right: Single runs. Top: GP-penalty when generating samples by perturbing training samples only. Middle: For GP-penalty, perturbing training and generated samples. Bottom: LP-penalty, perturbing training and generated samples (very similar to perturbing only training samples)

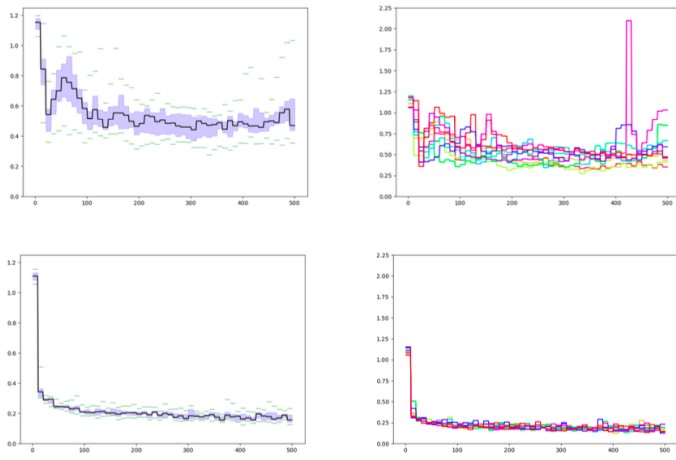

Figure 13: Evolution of the approximated EM distance during training of WGANs with local perturbation ($\lambda = 5$). Left: Median results over the 10 runs. Right: Single runs. Top: For the GP-penalty. Bottom: For the LP-penalty

## D.5 OPTIMIZING THE WASSERSTEIN-2 DISTANCE

We trained a WGAN with the objective of minimizing the Wasserstein-2 distance[8], that is, with the regularization term given by

$$\max\left(\left\{0, \frac{|f(x) - f(y)|}{||x - y||_2^2} - 1\right\}\right)^2 \ , \tag{11}$$

and penalty weight $\lambda = 10$. Results for the evolution of the critics loss and the approximated EM distance during training on the Swiss Roll data set are shown in Figure 14. Both critic loss and EM reduce smoothly, which makes the Wasserstein-2 distance (in combination with its theoretical properties) an interesting candidate to further investigations.

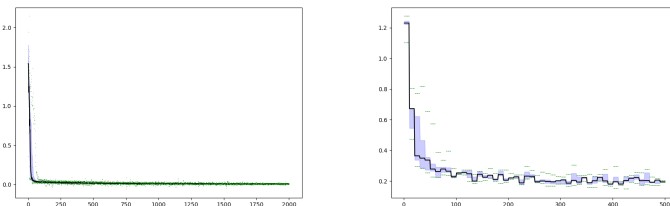

Figure 14: Evolution of the WGAN critics loss (Left) and the approximated EM distance (Right) for a WGAN-LP trained to minimize the Wasserstein-2 distance ($\lambda = 10$). Shown are the medians over 5 runs.

## D.6 EXPERIMENTAL RESULTS ON CIFAR

**Inception score.** The inception score was proposed by Salimans et al. (2016) to evaluate the quality of images $x$ sampled from a generative model $\nu$ based on the Inception model. Let $p(y|x)$ be the conditional probability of label $y$ for image $x$ under the Inception model and $p(y) = \int p(y|x)\nu(x)dx$ the marginal probability of labels $y$ with respect to samples generated from $\nu$. Then the Inception score is given by

$$\exp\left(\mathbb{E}_{x \sim \nu}[KL(p(y|x), p(y)]\right) \ . \tag{12}$$

Intuitively, a good generative model should produce samples for which the conditional label distribution has low entropy, while the variability over samples and thus the entropy of the marginal label distribution should be high. Therefore, a higher Inception score indicates a better performance of the generative model.

The maximal Inception scores reported in Table 1 are representative for the general evolution of the scores for WGAN-LP and WGAN-GP during training. As an example we show the evolution of the Inception score for penalty weights of $\lambda = 5$ and $\lambda = 100$ in Figure 15. It becomes clear that WGAN-GP performs similar to WGAN-LP for small values of the regularization parameter but much worse for larger values (this was consistently observed in all experiments). In Figure 16 we compare the performance of WGAN-LP and WGAN-GP in terms of the critics loss on a separate validation set, which again demonstrates a more stable behavior for WGAN-LP with respect to the choice of lambda.

---

[8]Choosing $p = 2$ has the theoretical advantage that, if the distributions $\mu$ and $\nu$ have finite moment of order 2 and are absolutely continuous with respect to the Lebesgue measure, then there is a unique deterministic optimal coupling.

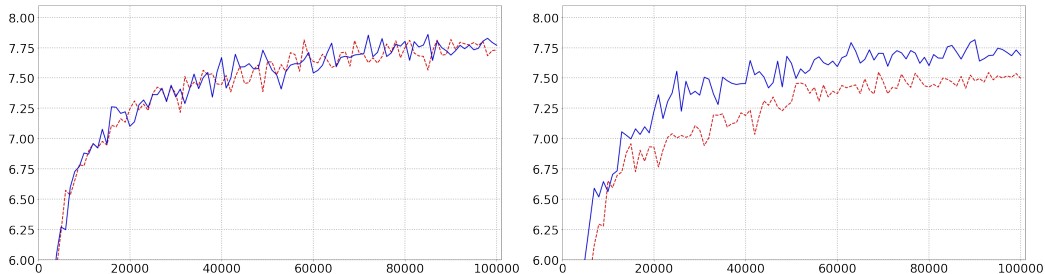

Figure 15: Evolution of Inception score on CIFAR for WGAN-LP in blue (solid) and WGAN-GP in red (dotted). Left: for regularization parameter $\lambda = 5$. Right: for regularization parameter $\lambda = 100$.

We also trained WGAN-GP and WGAN-LP with a conditional model (making use of the label information of CIFAR10) with $\lambda = 10$ and found a similar performance for both, i.e. $8.537 \pm 0.133$ and $8.462 \pm 0.115$ for WGAN-GP and WGAN-LP, respectively.

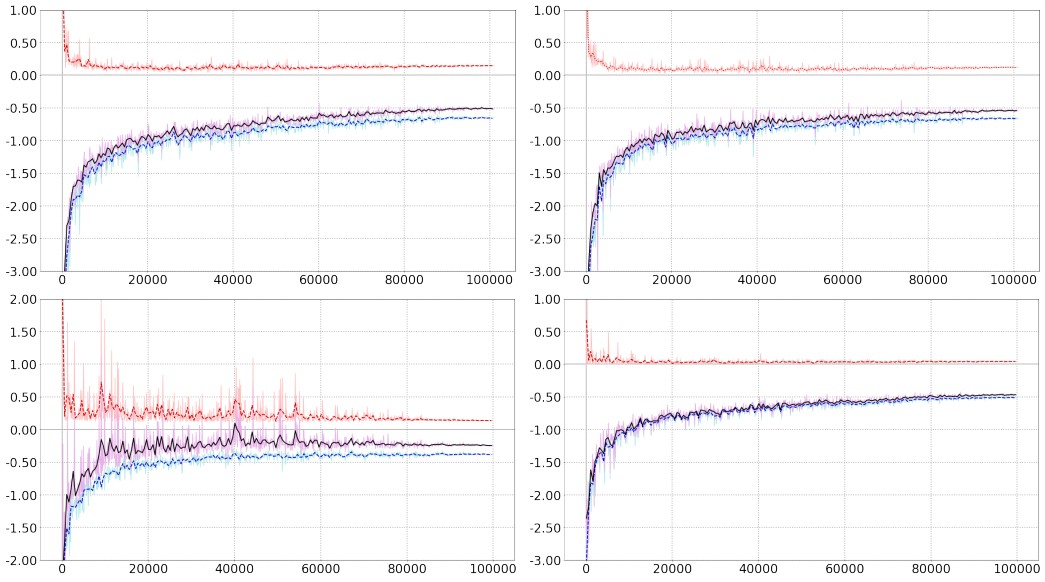

Figure 16: Evolution of validation loss on CIFAR. Black/purple curves indicate the total loss, blue curves the loss without regularization term, and red the regularization term only. Light colored curves indicate the true values, dark solid lines the average over a window of 5 iterations. Left: WGAN-GP. Right: WGAN-LP. Top: with $\lambda = 5$. Bottom: with $\lambda = 100$.

### D.7   RELATED PENALTIES

Level sets for WGANs trained with the regularization terms given by Equation (7) and (9) and penalty weight 10 are shown in Figure 17. As the evolution of the level sets and the sampled points indicate, training properly converges. However, on CIFAR-10, the same penalties did not lead to good results. As shown in Figure 18, using (7) for regularization initially lead to improving Inception scores but then quickly started to diverge, while using (9) lead to even greater instability.

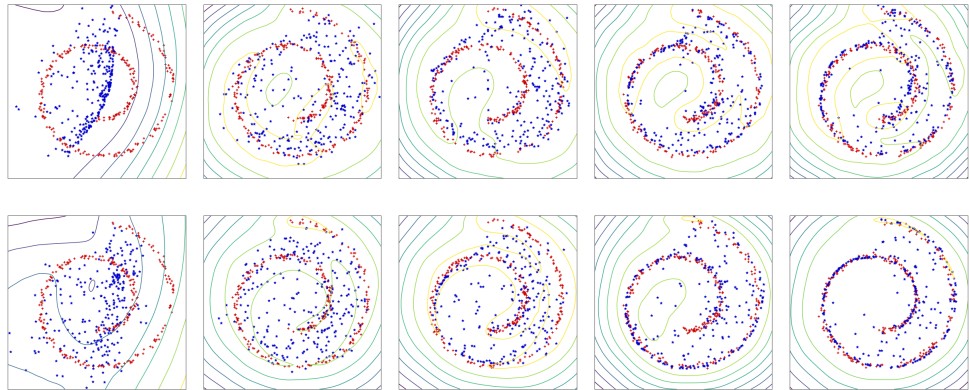

Figure 17: Level sets of the critic $f$ of WGANs during training, after 10, 50, 100, 500, and 1000 iterations. Yellow corresponds to high, purple to low values of $f$. Training samples are indicated in red, generated samples in blue. Top: With the regularization term given in Equation (7) and $\lambda = 10$. Bottom: With the regularization term given in Equation (9) and $\lambda = 10$.

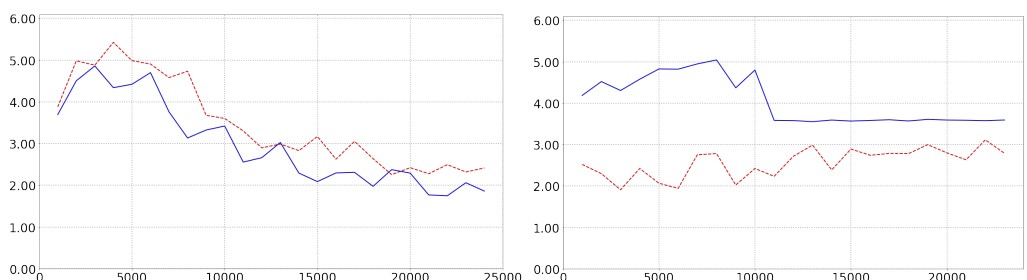

Figure 18: Inception scores for regularization Equation (7) for penalty weights 100 (red) and 5 (blue), shown on the left, and Inception scores for training with the regularization Equation (9) for penalty weights 100 (red) and 5 (blue), shown on the right.

