# OpenReview forum: "On the regularization of Wasserstein GANs"
_ICLR.cc/2018/Conference — Accept (Poster)_

### Official Review · AnonReviewer2 · 2017-11-26
**On the regularization of Wasserstein GANs**

**Rating:** 2
**Confidence:** 2

**Review:**

This paper is proposing a new formulation for regularization of Wasserstein Generative Adversarial models (WGAN). The original min/max formulation of the WGAN aim at minimizing over all measures, the maximal dispersion of expectation for 1-Lipschitz with the one provided by the empirical measure. This problem is often regularized by adding a "gradient penalty", \ie a  penalty of the form "\lambda E_{z~\tau}}(||\grad f (z)||-1)^2" where \tau is the distribution of (tx+(1-x)y) where x is drawn according to the empirical measure and y is drawn according to the target measure. In this work the authors consider substituting the previous penalty by "\lambda E_{z~\tau}}(max( ||\grad f (z)||-1,0)^2".

Overall the paper is too vague on the mathematical part, and the experiments provided are not particularly convincing in assessing the benefit of the new penalty.
The authors have tried to use mathematical formulations to motivate their choice, but they lack rigorous definitions/developments to make their point convincing.
They should also present early their model and their mathematical motivation: in what sense is their new penalty "preferable"?



Presentation issues:
- in printed black and white versions most figures are meaningless.
- red and green should be avoided on the same plots, as colorblind people will not perceived any difference...
- format for images should be vectorial (eps or pdf), not jpg or png...
- legend/sizes are not readable (especially in printed version).

References issues:
- harmonize citations: if you add first name for some authors add them for all of them: why writing Harold W. Kuhn and C. Vilani for instance?
- cramer->Cramer
- wasserstein->Wasserstein (2x)
- gans-> GANs
- Salimans et al. is provided twice, and the second is wrong anyway.



Specific comments:

page 1:
- "different more recent contributions" -> more recent contributions
- avoid double brackets "))"

page 2:
- Please rewrite the first sentence below Definition 1 in a meaningful way.
- Section 3: if \mu is an empirical distribution, it is customary to write it \mu_n or \hat \mu_n (in a way that emphasizes the number of observations available).
- d is used as a discriminator and then as a distance. This is confusing...

page 3:
- "f that plays the role of an appraiser (or critic)...": this paragraph could be extended and possibly elements of the appendix could be added here.
- Section 4: the way clipping is presented is totally unclear and vague. This should be improved.
- Eq (5): as written the distribution of \tilde{x}=tx+(1-t)y is meaningless: What is x and y in this context? please can you describe the distributions in a more precise way?
- Proof of Proposition 5 (cf. page 13): this is a sketch of proof to me. Please state precise results using mathematical formulation.
- "Observation 1": real and generated data points are not introduced at this stage... data points are not even introduced neither!

page 5:
- the examples are hard to understand. It would be helpful to add the value of \pi^* and f^* for both models, and explaining in details how they fit the authors model.
- in Figure 2 the left example is useless to me. It could be removed to focus more extensively on the continuous case (right example).
- the the -> the

page 6:
- deterministic coupling could be discussed/motivated when introduced. Observation 3 states some property of non non-deterministic coupling but the concept itself seems somehow to appear out of the blue.

page 10:
- Figure 6: this example should be more carefully described in terms of distribution, f*, etc.

page 14:
- Proposition 1: the proof could be shorten by simply stating in the proposition that f and g are distribution...

page 15:
- "we wish to compute"-> we aim at showing?
- f_1 is not defined sot the paragraph "the latter equation..." showing that almost surely x \leq y is unclear to me, so is the result then.
It could be also interesting to (geometrically) interpret the coupling proposed. The would help understanding the proof, and possibly reuse the same idea in different context.

page 16:
- proof of Proposition 2 : key idea here is using the positive and negative part of (f-g). This could simplify the proof.

---

> ### Author Response · Authors · 2018-01-05
> **Authors comments in reply to Reviewer2**
>
> We  thank the reviewer for the comments and suggestions and for checking the details of the arguments presented in the paper and thereby detecting room for substantial improvements.
>
> This led to the following changes in the revised version of our paper:
> We solved the issues in the reference section.
> We improved the presentation according to your suggestions whenever possible (as in proofs), improved the formulations, and removed typos.
> We improved the images. In particular, we would like to thank the reviewer for noticing the red/green issue that we missed to take care of in some plots. Our new images should thereby be better to read and understand.
>
> In the following we will reply directly to specific comments:
>
>
> >> “Overall the paper is too vague on the mathematical part, and the experiments provided are not particularly convincing in assessing the benefit of the new penalty.
> The authors have tried to use mathematical formulations to motivate their choice, but they lack rigorous definitions/developments to make their point convincing.”
>
> Unfortunately, the complaint about the lack of rigour is too broad for us to understand what exactly the reviewer is missing. We do believe, however, that the mathematical formulations are complete and concise, only due to the limited space available, we were forced to move most of the mathematical proofs into the appendix. We would be happy to improve by adding missing definitions or arguments that we are unaware of, if we get pointed to specific suggestions.
>
> >> “They should also present early their model and their mathematical motivation: in what sense is their new penalty "preferable"?”
>
> The new penalty is preferable over the previous ones, since
> The new penalty does not exclude approximations of optimal critic functions as the weight clipping approach does,
> does not enforce a constraint that cannot be justified,
> is therefore less dependent on the choice of the hyperparameter lambda,
> still builds on the great advantages of WGANs, which are one of the best performing GANs currently out there (and even leads to slightly better and more stable performance in practice).
>
> We made points 1,2 and 4 more clear in revised version of the paper and have added
> more theoretical results in Section 5 and experimental results in Section 6 to verify point 3.

---

> > ### Author Response · Authors · 2018-01-05
> > **Authors comments in reply to Reviewer2 - continued**
> >
> > >> Section 3: if \mu is an empirical distribution, it is customary to write it \mu_n or \hat \mu_n (in a way that emphasizes the number of observations available).
> >
> > The distributions do not necessarily need to be empirical here. Therefore, we decided to keep the distributions general with the according notation.
> >
> > >> page 3:
> > - "f that plays the role of an appraiser (or critic)...": this paragraph could be extended and possibly elements of the appendix could be added here.
> >
> > In a longer version, it would be nice to elaborate on this point in the main paragraph. Only due to the strong space constraints we were forced to move these considerations into the appendix, since (despite being helpful for understanding) they are not relevant for the rest of the paper.
> >
> > >> “- Section 4: the way clipping is presented is totally unclear and vague. This should be improved.”
> >
> > We state that weight clipping is “to enforce the parameters of the network not to exceed a certain value c_max>0 in absolute value”. Translated into formulas this is: there is some c_max>0 such that  |p|<c_max for all network parameters p, which is exactly the definition of weight clipping.
> >
> > >> “- Proof of Proposition 5 (cf. page 13): this is a sketch of proof to me. Please state precise results using mathematical formulation.”
> >
> > Unfortunately we are unsure what the reviewer is referring to. (There is no Prop 5 in the originally submitted version). We assume that Proposition 1 is meant.
> > In that case, we believe our version qualifies as more than a sketch of proof since it contains the complete set of arguments. If a reader feels more comfortable with mathematical notation, the reader may translate the written words into mathematical formulas to follow the arguments. In this case, we believe that formulas would even distract from the simplicity of arguments. In the end, our intention is to simplify the proof taken from the paper “Improving training of Wasserstein GANs” (https://arxiv.org/abs/1704.00028).
> >
> >
> > >> “page 5:”
> > - the examples are hard to understand. It would be helpful to add the value of \pi^* and f^* for both models, and explaining in details how they fit the authors model.
> >
> > We have added a labeled y-axis for the values of f*. The optimal coupling is indicated in red as before. In terms of (generalized) probability distributions, this would correspond to delta functions defined by the coupled points X and Os.
> >
> > >> “- in Figure 2 the left example is useless to me. It could be removed to focus more extensively on the continuous case (right example).”
> >
> > The discrete case is in our opinion much easier to understand than the continuous one, but motivates the reasoning for an optimal critic in the continuous case. We therefore suggest to leave it in. (see also our comments on the suggestion of a geometrical interpretation of the coupling in the proof to Proposition 1)
> >
> > >> “page 6:
> > - deterministic coupling could be discussed/motivated when introduced. Observation 3 states some property of non non-deterministic coupling but the concept itself seems somehow to appear out of the blue.”
> >
> > We have added a short discussion on deterministic couplings.

---

> > > ### Author Response · Authors · 2018-01-05
> > > **Authors comments in reply to Reviewer2 - continued**
> > >
> > >
> > > >> “page 10:
> > > - Figure 6: this example should be more carefully described in terms of distribution, f*, etc.”
> > >
> > > The optimal coupling is described in the text. Drawing the corresponding connections of the coupling would make the image less clear. The continuous function is also sufficiently described in the text by defining the slope almost everywhere, recalling that any choice of  y-intercept will produce an optimal critic function.
> > >
> > > >> “page 14:
> > > - Proposition 1: the proof could be shorten by simply stating in the proposition that f and g are distribution…”
> > >
> > >
> > > We suspect that this refers to Proposition 2 instead.
> > > In this case, we believe the proof is easier to phrase by starting with the density functions directly instead of starting with the distributions and then moving to the density functions (which we feel is necessary for our proof)
> > >
> > > page 15:
> > >
> > >
> > > >>“- f_1 is not defined sot the paragraph "the latter equation..." showing that almost surely x \leq y is unclear to me, so is the result then.”
> > >
> > > The latter equation was indeed unclear as written down. We have corrected it and removed f_1 from the notation.
> > >
> > > >>”It could be also interesting to (geometrically) interpret the coupling proposed. This would help understanding the proof, and possibly reuse the same idea in different context.”
> > >
> > > The geometric intuition is given by the discrete example from Figure 2. This is also exactly the reason why we suggest to keep the discrete case in the paper.
> > >
> > > In words that would be to move the left/right half of one distribution to the left/right half of the other distribution respectively. (We have added such a sentence to the beginning of the proof). We then use the freedom of non-uniqueness of the optimal coupling to simply find any coupling doing exactly that.
> > >
> > > >>” page 16:
> > > - proof of Proposition 2 : key idea here is using the positive and negative part of (f-g). This could simplify the proof.”
> > >
> > > We do not quite understand this comment, as using the positive and negative part of f-g is exactly what we are doing. We have added the comment that the mathematical formulas describe exactly the positive and the negative part of (f-g).

---

### Official Review · AnonReviewer3 · 2017-11-27
**Nice regularization term for WGAN with strong relation to regularized OT**

**Rating:** 7
**Confidence:** 4

**Review:**

This paper proposes a novel regularization scheme for Wasserstein GAN based on a relaxation of the constraints on the Lipschitz constant of 1. The proposed regularization penalize the critic function only when its gradient has a norm larger than one using some kind of squared hinge loss. The reasons for this choice are discussed and linked to theoretical properties of OT. Numerical experiments suggests that the proposed regularization leads to better posed optimization problem and even a slight advantage in terms of inception score on the CIFAR-10 dataset.

The paper is interesting and well written, the proposed regularization makes sens since it is basically a relaxation of the constraints and the numerical experiments also suggest it's a good idea. Still as discussed below the justification do not address a lots of interesting developments and implications of the method and should better discuss the relation with regularized optimal transport.

Discussion:

+ The paper spends a lot of time justifying the proposed method by discussing the limits of the "Improved training of Wasserstein GAN" from Gulrajani et al. (2017). The two limits (sampling from marginals instead of optimal coupling and differentiability of the critic) are interesting and indeed suggest that one can do better but the examples and observations are well known in OT and do not require proof in appendix. The reviewer believes that this space could be better spend discussing the theoretical implication of the proposed regularization (see next).

+ The proposed approach is a relaxation of the constraints on the dual variable for the OT problem. As a matter of fact we can clearly recognize a squared hinge loss is the proposed loss. This approach (relaxing a strong constraint) has been used for years when learning support vector machines and ranking and a small discussion or at least reference to those venerable methods would position the paper on a bigger picture.

+ The paper is rather vague on the reason to go from Eq. (6) to Eq. (7). (gradient approximation between samples to gradient on samples). Does it lead to better stability to choose one or the other?
 How is it implemented in practice? recent NN toolbox can easily compute the exact gradient and use it for the penalization but this is not clearly discussed even in appendix. Numerical experiments comparing the two implementation or at least a discussion is necessary.

+ The proposed approach has a very strong relations to the recently proposed regularized OT (see [1] for a long list of regularizations) and more precisely to the euclidean regularization. I understand that GANS (and Wasserstein GAN) is a relatively young community and that references list can be short but their is a large number of papers discussing regularized optimal transport and how the resulting problems are easier to solve. A discussion of the links is necessary and will clearly bring more theoretical ground to the method. Note that a square euclidean regularization leads to a regularization term in the dual of the form max(0,f(x)+f(y)-|x-y|)^2 that is very similar to the proposed regularization. In other words the authors propose to do regularized OT (possibly with a new regularization term) and should discuss that.

+ The numerical experiments are encouraging but a bit short. The 2D example seem to work very well and the convergence curves are far better with the proposed regularization. But the real data CIFAR experiments are much less detailed with only a final inception score (very similar to the competing method) and no images even in appendix. The authors should also define (maybe in appendix) the conditional and unconditional inception scores and why they are important (and why only some of them are computed in Table 1).

+ This is more of a suggestion. The comparison of the dual critic to the true Wasserstein distance is very interesting. It would be nice to see the behavior for different values of lambda.


[1] Dessein, A., Papadakis, N., & Rouas, J. L. (2016). Regularized Optimal Transport and the Rot Mover's Distance. arXiv preprint arXiv:1610.06447.


Review update after reply:

The authors have responded to most of my concerns and I think the paper is much stronger now and discuss the relation with regularized OT. I change the rating to Accept.

---

> ### Author Response · Authors · 2018-01-05
> **Authors comments in reply to Reviewer3**
>
> We thank the reviewer for his highly valuable comments and thoughtful suggestions! Based on them, we applied the following main changes in the revised version of our paper:
> We added a paragraph giving a short introduction to regularized OT in Section 2 and a paragraph about the connection to our proposed regularization in Section 5  (special thanks for pointing us in this direction!!!).
> We extended the CIFAR experiments, by running more experiments with different  values of the regularization parameter (all show that WGAN-LP produces equivalent or better results and is less sensitive to the value of the regularization parameter) and presenting a deeper investigation of the loss contributions of the regularization term. Interestingly we find, that the penalty of WGAN-GP is behaving similar to the one of WGAN-LP in settings with low regularization parameter. We have added theoretical considerations explaining this behaviour in Section 5.
>
> In the following we will reply directly to specific comments:
>
> >> “The paper spends a lot of time justifying the proposed method by discussing the limits of the "Improved training of Wasserstein GAN" from Gulrajani et al. (2017). The two limits (sampling from marginals instead of optimal coupling and differentiability of the critic) are interesting and indeed suggest that one can do better but the examples and observations are well known in OT and do not require proof in appendix. The reviewer believes that this space could be better spend discussing the theoretical implication of the proposed regularization (see next).”
>
> We haven’t been able to find references, where computations of the examples can be found in the literature. Approaching WGANs from a deep learning viewpoint, we are also convinced that researchers interested in GANs without the necessary background in OT will find a quick discussion of the examples at least very helpful but possibly even necessary. (See also opposing comments by Reviewer 2.) We have moved as much as we believe is adequate to the appendix.
>
> “The proposed approach is a relaxation of the constraints on the dual variable for the OT problem. As a matter of fact we can clearly recognize a squared hinge loss is the proposed loss. This approach (relaxing a strong constraint) has been used for years when learning support vector machines and ranking and a small discussion or at least reference to those venerable methods would position the paper on a bigger picture.”
>
> We added a sentence referring to relaxation of hard constraints in the objective of SVMs.
>
> >>” The paper is rather vague on the reason to go from Eq. (6) to Eq. (7). (gradient approximation between samples to gradient on samples). Does it lead to better stability to choose one or the other? How is it implemented in practice? recent NN toolbox can easily compute the exact gradient and use it for the penalization but this is not clearly discussed even in appendix. Numerical experiments comparing the two implementation or at least a discussion is necessary.”
>
> The main reason to go from  Eq. (6) to Eq. (7) is that enforcing the constraint on the gradient norm implements a valid constraint into all directions from the given point, not just a condition on the difference between two points (and just in only one direction). This should help for better generalization to unseen samples. We performed experiments to verify this (the results are shown in Appendix D in the revised version of the paper): While regularization based on Eq. (6)  worked well on toy data, it performed considerably weaker on CIFAR10, supporting the advantage of a regularization as given in Eq. (7).
>
> For the computation we did indeed use standard implementations of the gradient in tensorflow (see https://www.tensorflow.org/api_docs/python/tf/gradients and http://pytorch.org/docs/master/autograd.html#torch.autograd.grad).
> Links to our code will be provided in case of acceptance.
>
> .

---

> > ### Author Response · Authors · 2018-01-05
> > **Authors comments in reply to Reviewer3-continued**
> >
> > >> “+ The proposed approach has a very strong relations to the recently proposed regularized OT (see [1] for a long list of regularizations) and more precisely to the euclidean regularization. I understand that GANS (and Wasserstein GAN) is a relatively young community and that references list can be short but their is a large number of papers discussing regularized optimal transport and how the resulting problems are easier to solve. A discussion of the links is necessary and will clearly bring more theoretical ground to the method. Note that a square euclidean regularization leads to a regularization term in the dual of the form max(0,f(x)+f(y)-|x-y|)^2 that is very similar to the proposed regularization. In other words the authors propose to do regularized OT (possibly with a new regularization term) and should discuss that.”
> >
> > We are very thankful for pointing us to the link to regularized OT. The similarity to regularized OT with Euclidean regularization is highly interesting, and we discuss it now in Sections 2 and 5.
> >
> > But, at least from our understanding, the equivalence to regularized OT is not exactly given, since a regularization term in the primal does not seem to allow for the maximization in the dual over one function only. We believe the same problem to appear for any Bregman divergence and therefore doubt that any new regularization term gives the exact equivalence to any of the previously considered approaches in regularized OT that we are aware of.
> > We performed experiments with the variant of our regularization term, that is most similar to the Euclidean regularized OT (see last paragraph in the experimental section), but it showed only good performance on toy dataset, but poor performance on larger datasets such as CIFAR-10.
> >
> > >>+ The numerical experiments are encouraging but a bit short. The 2D example seem to work very well and the convergence curves are far better with the proposed regularization. But the real data CIFAR experiments are much less detailed with only a final inception score (very similar to the competing method) and no images even in appendix.
> >
> > We run additional experiments on CIFAR-10 for 3 more values of lambda (0.1,5,100), all supporting the conclusion that  WGAN-LP performs slightly better is much less dependent on the right choice of hyperparameter lambda than the  WGAN-GP (see Table 1 in the revised version). We also added a plot  (Fig 6) displaying the regularization term of WGAN-GP separated into contributions based on a gradient norm exceeding one and based on a gradient norm smaller one, which also supports the higher sensitivity of WGAN-GP to the right choice of hyperparameter, and additionally suggests that WGAN-GP in fact behaves similar to WGAN-LP  when the hyperparameter lambda is chosen small enough to make it perform well.
> >
> > >> “The authors should also define (maybe in appendix) the conditional and unconditional inception scores and why they are important (and why only some of them are computed in Table 1)”
> >
> > We added such a description into Appendix D.6.
> >
> > >>”This is more of a suggestion. The comparison of the dual critic to the true Wasserstein distance is very interesting. It would be nice to see the behavior for different values of lambda.”
> >
> > Due to limitations in our access to computational resources, we were not yet able to conduct these experiments, but agree that this would be very interesting and plan to report such results in the camera ready version

---

### Official Review · AnonReviewer1 · 2017-11-30
**A too preliminary discussion of ways of regularizing Gans with the L_1 Wasserstein metric**

**Rating:** 6
**Confidence:** 5

**Review:**

The article deals with regularization/penalization in the fitting of GANs, when based on a L_1 Wasserstein metric. Basics on mass transportation are briefly recalled in section 2, while section 3 formulate the GANs approach in the Wasserstein context. Taking into account the Lipschitz constraint and (non-) differentiability of optimal critic functions f are discussed in section 4 and Section 5 proposes a way to penalize candidate functions f that do not satisfy the Lipschitz condition using a tuning parameter lambda, ruling a trade-off between marginal fitting and gradient control. The approach is illustrated by numerical experiments. Such results are hardly convincing, since the tuning of the parameter lambda plays a crucial role in the performance of the method. More importantly, The heuristic proposed in the paper is interesting and promising in some respects but there is a real lack of theoretical guarantees motivating the penalty form chosen, such a theoretical development could allow to understand what may rule the choice of an ideal value for lambda in particular.

---

> ### Author Response · Authors · 2018-01-05
> **Authors comments in reply to Reviewer1**
>
> We thank the reviewer for the valuable feedback.
>
> We share the viewpoint that theoretical guarantees would be very much desirable and should further investigated, however we also think that rigorous convergence results, as for example in convex optimization, are hard to establish in a field of deep learning approaches, where there is still the lack of theoretical understanding in general.
> On the other hand, we do believe that our research provides sufficient theoretical evidence for our method to be advantageous over existing approaches to WGANs.
>
> In the following we will reply directly to specific comments:
>
> >> “The approach is illustrated by numerical experiments. Such results are hardly convincing, since the tuning of the parameter lambda plays a crucial role in the performance of the method.”
>
> It is a weakness of many models that they do depend on tuning hyperparameters in a very sensitive way. This has also been demonstrated for various GANs in a recent paper (https://arxiv.org/abs/1711.10337). Our results, however, demonstrate that our version, WGAN-LP,  is less sensitive to the tuning of lambda than WGAN-GP.  That alone is a big advantage of our version to existing ones in our opinion. We tried to make this point more clear in the revised version and added theoretical considerations and more experimental results on CIFAR with different choices of the hyperparameter which consistently show a better performance of WGAN-LP and less sensitivity to the right choice of lambda.
>
> >> “More importantly, the heuristic proposed in the paper is interesting and promising in some respects but there is a real lack of theoretical guarantees motivating the penalty form chosen, such a theoretical development could allow to understand what may rule the choice of an ideal value for lambda in particular.”
>
> We believe that our approach is theoretically justified in the sense that it does point out theoretical issues of former approaches that were not noticed and corrects them. In this way it improves on one of best-working GANs in a theoretically justified way.
> In the revised version, we are now also discussing the link to regularized optimal transport theory (see new paragraphs in Sections 2 and 5).
> We agree, that a theoretical analysis that could guide the choice of the right value of the hyperparameter would be highly desirable, but those guarantees are hard to derive.
> From a theoretical viewpoint (that we could not see reflected in experimental results however) we believe high hyperparameter choices would be ideal, since they “strengthen” the weak constraint. A choice of a high value for lambda  would also be justified by the newly added connection to optimal transport theory (see Section 5). In addition, we added some theoretical observations on the dependence on lambda in Section 5.

---

### Public Comment · ~Junghoon_Seo1 · 2017-12-16
**Reproduction report of this paper**

We decided to reproduce the experiment in this paper to participate in ICLR 2018 Reproducibility Challenge. Reports are currently submitted to arXiv and pending. Maybe it will be released after 2-3 days. The temporary submission
identifier is: submit/2105336. Below is a summary of the report. The reproduction code is in https://github.com/mikigom/WGAN-LP-tensorflow.

The ultimate goal of all experiments in this paper is that WGAN-GP is better than Equation WGAN-LP in aspect of training stability and convergence speed to optimize optimal transport problem in GAN framework.

Note that subsection 'Sample quality on CIFAR-10' in the experiment section of this paper and Appendix D.5 'Optimizing the Wasserstein-2 distance' are out of our reproduction scope. At the beginning of the reproduction, we refer to the arXiv uploaded version of the paper. However, we noticed that the revised version of OpenReview recently added subsection 'Sample quality on CIFAR-10'. As a matter of time, the experiments in this subsection were excluded from the scope of this report. On the other hand, we are having trouble implementing a regularization loss term that minimizes the Wasserstein-2 distance in Tensorflow.

The reproduction code consists of four Python modules: data_generator.py, model.py, reg_losses.py, and trainer.py. data_generator.py is a module that provides a class that generates the sample data needed for learning. model.py is a module that implements 3-layer neural networks for a generator and a critic. reg_losses.py defines the sampling method and loss term for regularization. trainer.py includes a pipeline for model learning and visualization. To see our implementation in more detail, please check out the repository.

We first want to specify that the experiment follows the trends presented in the paper as a whole, but the overall learning speed is relatively slow. It is assumed that this is due to differences of hyperparameter in RMSProp, or in unrecognized elements. However, since this is not very inconsistent with the overall tendency of the experiment, we proceeded to reproduce the experiments without searching to solve it. It takes about 12 minutes to learn the 20k steps without EMD calculation, and it takes about 2 hours to learn the 2k steps when EMD calculation is included.

Please refer to the report to be published on arXiv for detailed results of the experiment.

We have confirmed what the target papers claimed: First, WGAN-LP has more stable learning and faster convergence property than WGAN-GP. Second, WGAN-LP is much more robust to regularization fraction λ than WGAN-GP. Finding and determining the appropriate hyperparameter is an important but cumbersome, on study of machine learning. Therefore, presenting a robust model to the selection of hyperparameters can be a sufficient contribution to other researchers and the field itself. We are able to accept that the target paper contributes to this part in a reproducible way.

-- Update on December 20 2017
Our report is completely uploaded on arXiv.
url : https://arxiv.org/abs/1712.05882

---

> ### Author Response · Authors · 2018-01-05
> **Authors reply to reproduction team**
>
> Thank you very much for your interest in our work. We are happy to hear that you could reproduce and confirm our results. We wish you all the best for the ICLR 2018 Reproducibility Challenge!

---

### Decision · Program_Chairs · 2018-01-29
**ICLR 2018 Conference Acceptance Decision**

**Decision:**

Accept (Poster)

**Comment:**

This paper proposes an interesting analysis of the limitations of WGANs as well as a solution to these limitations. I am not too convinced by the experimental part as, as some of the reviewers have mentioned, it relies on hyperparameters which can be hard to tune.

The more theoretical part, even if it could be written with more care as pointed out by reviewer 2, is nonetheless interesting and could stir discussion. I think it would be a good addition to ICLR as a poster.